# pH-dependence of the *Plasmodium falciparum* chloroquine resistance transporter is linked to the transport cycle

Fiona Berger[1], Guillermo M. Gomez [1], Cecilia P. Sanchez[1], Britta Posch[1], Gabrielle Planelles[2], Farzin Sohraby[3], Ariane Nunes-Alves [3] ✉ & Michael Lanzer [1] ✉

The chloroquine resistance transporter, PfCRT, of the human malaria parasite *Plasmodium falciparum* is sensitive to acidic pH. Consequently, PfCRT operates at 60% of its maximal drug transport activity at the pH of 5.2 of the digestive vacuole, a proteolytic organelle from which PfCRT expels drugs interfering with heme detoxification. Here we show by alanine-scanning mutagenesis that E207 is critical for pH sensing. The E207A mutation abrogates pH-sensitivity, while preserving drug substrate specificity. Substituting E207 with Asp or His, but not other amino acids, restores pH-sensitivity. Molecular dynamics simulations and kinetics analyses suggest an allosteric binding model in which PfCRT can accept both protons and chloroquine in a partial noncompetitive manner, with increased proton concentrations decreasing drug transport. Further simulations reveal that E207 relocates from a peripheral to an engaged location during the transport cycle, forming a salt bridge with residue K80. We propose that the ionized carboxyl group of E207 acts as a hydrogen acceptor, facilitating transport cycle progression, with pH sensing as a by-product.

The chloroquine resistance transporter, PfCRT, is a drug/metabolite carrier with a surprisingly broad substrate specificity and a critical role in the physiology and pathophysiology of the human malaria parasite *Plasmodium falciparum* during intraerythrocytic development. PfCRT resides at the membrane of the digestive vacuole[1], where it acts as a solute carrier for oligopeptides[2,3]. These oligopeptides are structurally diverse and 4–11 residues in length[2,3]. They originate from hemoglobin and other red blood cell proteins, which the intracellular parasite ingests and subsequently digests in its food vacuole[3–5]. The transport of oligopeptides into the cytoplasm of the parasite feeds into the metabolic circuits of the parasite and, in addition, maintains the colloidal osmotic balance of the infected erythrocytes in light of the anabolic and space-demanding activities of the parasite[6–8].

Several antimalarial drugs, including the former first-line drug chloroquine (CQ) and related currently deployed quinoline and quinoline-like antimalarials, such as piperaquine (PPQ), quinine (QN) and amodiaquine, target the digestive vacuole by inhibiting the detoxification of heme, the prosthetic group of hemoglobin, to inert hemozoin[9,10]. Resistance to quinolines and structurally unrelated compounds is mediated by PfCRT[11–18], which after acquisition of 4 to 10 mutational changes, expels these drugs from the digestive vacuole and, hence, away from their molecular target[19–21].

While our understanding of the substrate specificity of PfCRT and the mutational changes converting PfCRT into a drug-transporting system has increased in recent years, central questions regarding the transport process itself have remained unresolved. For example, the

[1]Center of Infectious Diseases, Parasitology, Universitätsklinikum Heidelberg, Im Neuenheimer Feld 324, 69120 Heidelberg, Germany. [2]INSERM, Centre de Recherche des Cordeliers, Unité 1138, CNRS ERL8228, Université Pierre et Marie Curie and Université Paris-Descartes, Paris 75006, France. [3]Institute of Chemistry, Technische Universität Berlin, Straße des 17. Juni 135, 10623 Berlin, Germany. ✉e-mail: ferreira.nunes.alves@tu-berlin.de; michael.lanzer@med.uni-heidelberg.de

transport of oligopeptides and CQ is pH sensitive[3,22]. In the case of CQ, the transport activity reaches a maximum at a pH of ~6.0[22]. As a consequence, PfCRT operates at only 60% of its full transport velocity at the pH of the digestive vacuole of 5.2[23]. Why the strong selective forces on PfCRT have not been able to shift the pH optimum is unclear as is the structural basis for the observed pH sensitivity. Furthermore, previous studies have shown that PfCRT-mediated CQ transport is electrogenic[24] and associated with a proton leak from the digestive vacuole[25,26]. These findings led to the hypothesis of PfCRT functioning as a proton-coupled symporter. Whether the pH sensitivity of PfCRT and the proposed proton symport are related or unrelated events has not yet been addressed.

The previously reported structure of PfCRT at 3.2 Å resolution revealed a pseudo-symmetric and inverted antiparallel arrangement of the 10 transmembrane domains (TM) and a large negatively net-charged central cavity formed by TM1-TM4 and TM6-TM9[27]. Structural information regarding pH sensitivity or proton coupling was not provided.

Here we describe a systematic mutational analysis of PfCRT with a focus on proton-accepting residues. We found that a single residue, E207, serves as a critical pH sensor. Molecular dynamics (MD) simulations and kinetic studies suggest that E207 plays a pivotal role in the transport cycle, by serving as a proton acceptor for interactions with K80 via a salt bridge, thereby accelerating progression through the transport cycle. Protonation of E207 is proposed to interfere chemically and sterically with its role in the transport cycle.

## Results

### Identification of E207 as a critical pH sensor

We initially re-examined the pH dependence of PfCRT, by expressing the PfCRT variant of the CQ resistant *P. falciparum* strain Dd2 in *Xenopus laevis* oocytes, and determined the uptake of CQ from the external medium (50 μM unlabeled CQ and 42 nM tritiated CQ) at six different pH values varying from pH 4.5 to 7.0 in increments of 0.5 pH units (Fig. 1a). The resulting data confirmed the strong pH sensitivity of the CQ transport activity of PfCRT, which followed a bell-shaped curve with a maximum at pH $6.2 \pm 0.3$ (Fig. 1a). The lowest transport activity (24% of the maximal value) was observed at pH 4.5 (Fig. 1a, b). The inflection point in the weakly acidic pH range occurred at ~5.2, which might indicate the p$K_a$ value of the pH sensing group, assuming a simple model that does not consider the effect of the microenvironment or the possibility of several pH sensing groups contributing to the pH sensitivity.

Alternatively, the loss of transport activity at low pH may result from acid-induced damage to the protein, a process that is mostly irreversible[28]. To test this hypothesis, we compared the CQ transport activity of PfCRT$^{Dd2}$-expressing oocytes pre-treated for 1 h in medium with a pH of 4.5 with that of untreated PfCRT$^{Dd2}$-expressing oocytes and found no significant differences (Supplementary Fig. 1). This finding is consistent with the acid sensitivity of PfCRT being a physiological process rather than the result of a damaged secondary or tertiary structure. We also considered the possibility of CQ, the substrate of PfCRT, serving as the pH sensor. However, such a model appears unlikely. Although CQ is a diprotic weak base, there is virtually no change in the level of protonation between pH 4.5 and 6.0, with >99% of the molecules being di-protonated, given the p$K_a$ values of CQ of 8.1 and 10.2 for the quinoline nitrogen and the aliphatic, tertiary amine, respectively[29].

To further explore the pH sensitivity of PfCRT, we systematically replaced potentially pH sensing residues within the sequence of PfCRT by alanines or, in the case of Asp contained within transmembrane domains by the isosteric amino acid Asn (Supplementary Fig. 2). We particularly focused on two classes of pH sensing amino acids: (i) the acidic amino acids Glu, Asp, and His, which have side chain p$K_a$ values of 4.25, 3.65, and 6.00, respectively[30], and which can be protonated or

deprotonated within the operative pH range of PfCRT; and (ii) the aromatic amino acids Phe, Tyr, and Trp, which can engage in cation–π interactions with protons through the polarized electron cloud of their aromatic side chains. The other amino acids are unable to sense pH changes within the operative pH range of PfCRT$^{Dd2}$ or, in the case of Ser, Thr, and Cys, can do so only under rare circumstances[31].

A total of 53 mutants (39 single amino acid substitutions and 2 constructs containing 7 amino acid substitutions each) were generated and functionally analyzed in the *X. laevis* oocyte system under standardized conditions (Supplementary Table 1), i.e., the amount of the CQ uptake was determined from an extracellular concentration of 50 μM after 1 h of incubation in medium with pH 4.5 or 6.0. To compare the data, an *R*-value[31,32] was calculated for each mutant by dividing the transport rate at pH 4.5 by that at pH 6.0 (Fig. 1b, c). The *R*-value can vary between 0 and 1 and is independent of the expression level. The *R*-value for PfCRT$^{Dd2}$ was ~0.2, indicative of a high pH sensitivity (Fig. 1b, c). pH-independent mutants are expected to have high *R*-values, but not necessarily equal to 1, since the pH can also affect other residues involved in the transport process[31,32].

The *R*-values were subsequently plotted against the corresponding CQ transport activity at pH 6.0 (Fig. 1d). Most of the mutants maintained some degree of pH sensitivity as indicated by an *R*-value between 0.25 and 0.6. Only one mutant, namely the E207A variant, stood out. E207A had an *R*-value of 0.9, while keeping a reasonable 44% of the transport activity at pH 6.0 compared with PfCRT$^{Dd2}$. Interestingly, the E208A mutation, which is located next to E207, displayed an *R*-value of 0.48, indicative of a pH sensitive carrier (Fig. 1d). The mutant D310A also had a high *R*-value of 0.77, but was not further considered because of its low transport activity at pH 6.0 of 1.25 pmol h$^{-1}$ oocyte$^{-1}$. In comparison, the mutant H97A displayed a higher transport activity at pH 6.0 than did PfCRT$^{Dd2}$, while maintaining a low *R*-value of 0.40. An extended investigation of E207A over the pH range from 4.5 to 7.0, confirmed that the CQ transport activity is pH insensitive (Fig. 1a). We noted one mutant, namely H273A, with an *R*-value smaller than that of PfCRT$^{Dd2}$. A small *R*-value results from a greater difference between the transport activities at pH 4.5 and 6.0 and indicates a mutant with increased pH sensitivity. Since H273A was pH dependent, it was not further investigated (Supplementary Fig. 3).

We next explored whether the pH sensitivity of PfCRT$^{Dd2}$ and the pH-insensitivity of E207A extended to drugs other than CQ. To this end, we performed the transport assay at pH 4.5 and 6.0, using radiolabeled QN, quinidine (QD) or PPQ (all at 50 μM external concentration), and analyzed the resulting *R*-values as a function of the transport activity at pH 6.0. As seen in Fig. 1d, E207A consistently had *R*-values that were 2.1–4.5-fold higher than those of PfCRT$^{Dd2}$, although the actual *R*-value varied depending on the drug investigated. In the case of PPQ, the *R*-value could only be approximated since the pKa values of PPQ are in the operative range of PfCRT (5.39, 5.72, 6.24, 6.88)[33]. As a consequence, the distribution of the various PPQ species likely differs between pH 4.5 and pH 6.0, with an impact on the transport activity, assuming distinct kinetic properties for each species.

Even though PfCRT$^{Dd2}$ and E207A differed with regard to their pH responses, they maintained comparable relative transport activities at pH 6.0 to the various drugs, as shown after normalization of the respective transport rates to that of CQ (Fig. 1e). Moreover, verapamil, a partial mixed type inhibitor[34] blocked CQ transport in both PfCRT$^{Dd2}$ and E207A (Fig. 1e). Neither PfCRT$^{Dd2}$ nor E207A acted on the anticancer drugs, vinblastine and vincristine (Fig. 1e). These findings suggest that the E207A mutation did not cause nonspecific structural changes that affect the substrate specificity of the carrier. Furthermore, the pH sensitivity of PfCRT$^{Dd2}$ appeared to be an intrinsic property of the transporter, with E207 playing a pivotal role in pH sensing.

E207 is present in all published PfCRT isoforms and it is also conserved among most *Plasmodium* species, with the exception of the

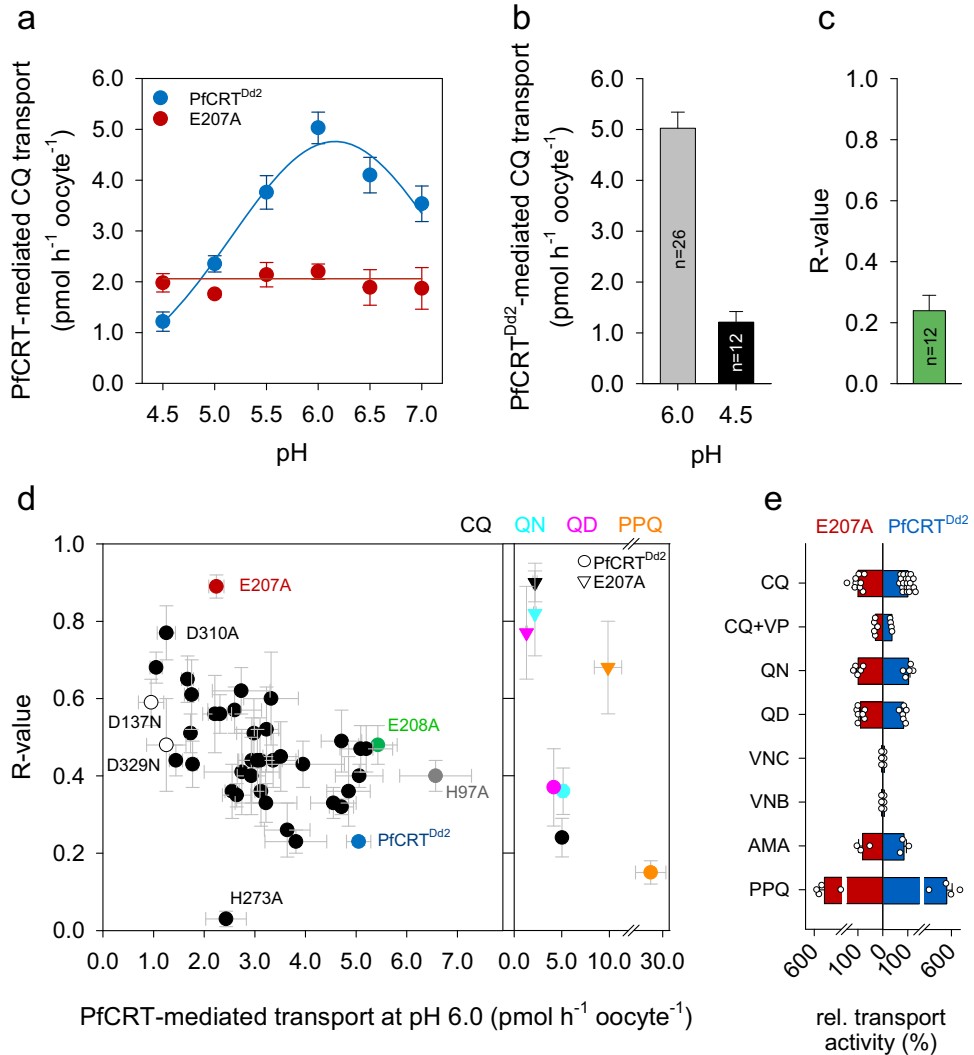

**Fig. 1 | I pH dependence of PfCRT$^{Dd2}$. a** CQ transport activity of PfCRT$^{Dd2}$ (blue symbols) and E207A (red symbols) as a function of the pH. Specific CQ transport was determined from an extracellular medium containing 50 μM of CQ. Each data point represents the mean ± SEM of n biologically independent samples, with $n$ being from left to right for PfCRT$^{Dd2}$: 12, 11, 13, 26, 12, and 15; and E207A: 9, 3, 3, 9, 3, and 3. **b** PfCRT$^{Dd2}$-mediated CQ transport activity at a pH of 6.0 and 4.5. The number of biologically independent samples ($n$) are indicated. **c** $R$-value of PfCRT$^{Dd2}$, as defined by the ratio of the CQ transport activity at pH 4.5 over that at pH 6.0. **d** *Left*, Identification of E207 as a critical pH sensor in PfCRT. Shown is a plot of $R$-values against PfCRT-mediated CQ uptake at pH 6.0 of 53 PfCRT$^{Dd2}$-derived mutants, in which titratable and aromatic, π-interacting residues were replaced by alanine or asparagine in the case of aspartic acid. Relevant mutants are highlighted. For further information on the mutants see Supplementary Table 1. *Right*, Shown is a plot of $R$-values against transport activity at pH 6.0 for PfCRT$^{Dd2}$ (circle) and E207A (triangle) with the substrates, CQ (black), quinine (QN, light blue), quinidine (QD, purple), and piperaquine (PPQ, orange). Each data point represents the mean ± SEM of 3 to 9 biologically independent samples, with the number of biologically independent sample varying from mutant to mutant. **e** Substrate selectivity of PfCRT$^{Dd2}$ and E207A. The transport activities were normalized to that of CQ at pH 6.0. Verapamil (VP), vincristine (VNC), vinblastine (VNB), amantadine (AMA). Data were analyzed using bar charts, with the mean ± SEM of $n$ biologically independent samples being shown, with n being from top to bottom for E207A: 9, 5, 5, 7, 3, 3, 3, and 4; and for PfCRT$^{Dd2}$: 26, 4, 5, 6, 3, 3, 3, and 4. The bar charts were overlaid with the individual data points. Source data for all subfigures are provided as a Source Data file[86].

four rodent malaria parasites, *P. berghei, P. chabaudi, P. vinckei* and *P. yoelii* (according to www.plasmodb.org as of April 2023) (Supplementary Fig. 4a). Interestingly, the CRT orthologues of the rodent malaria parasites cluster far away from PfCRT in a bootstrap analysis (Supplementary Fig. 4b), suggesting divergent physiological functions and, possibly, mechanisms of pH-regulation.

Even though the $R$-value is a dimensionless parameter independent of the expression level, we nevertheless investigated the expression of E207A in the oocyte system relative to that of PfCRT$^{Dd2}$. Immuno-fluorescence assays, using a specific antiserum to PfCRT, located both PfCRT$^{Dd2}$ and E207A at the oolemma as shown by colocalization of the fluorescence signal with that of Alexa 633-conjugated wheat germ agglutinin, a marker for the plasma

membrane of the oocyte[35] (Fig. 2a). Furthermore, semi-quantitative Western analyses revealed comparable expression levels of PfCRT$^{Dd2}$ and E207A relative to the internal standard α-tubulin (Fig. 2b, c). Water-injected oocytes analyzed in parallel did not react with the PfCRT specific antiserum.

## E207 protrudes into the substrate cavity
E207 is located in the digestive vacuolar loop between transmembrane domains 5 and 6. A structural model of PfCRT$^{Dd2}$ based on the published cryo-EM structure of the related, CQ resistance-mediating PfCRT$^{7G8}$ variant in the open-to-vacuole conformation[27] (Supplementary Fig. 5) revealed that E207 lies at the entrance of the substrate binding cavity, with its carboxyl side chain protruding laterally into the

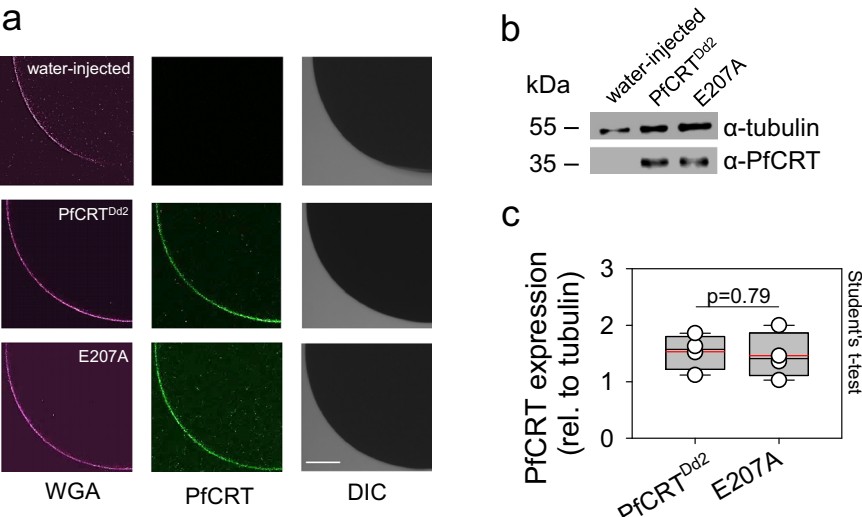

**Fig. 2 | I Expression of PfCRT variants in *X. laevis* oocytes. a** Confocal fluorescence images of fixed water-injected oocytes and PfCRT[Dd2] and E207A-expressing oocytes. *Left*, fluorescence image of Alexa 633 conjugated wheat germ agglutinin (WGA) as a marker of the oolemma. *Middle*, fluorescence image of PfCRT, using a specific rabbit antiserum and an anti-rabbit Alexa Fluor 546 secondary antibody. *Right*, differential interference contrast image (DIC). Scale bar, 135 μm. Representative images of 4 biologically independent samples. **b** Western analysis of total protein lysates from oocytes injected with water, PfCRT[Dd2]-RNA, or E207A-RNA, using a polyclonal guinea pig antiserum specific to PfCRT and a polyclonal rabbit antiserum specific to α-tubulin. Representative images of 4 biologically independent samples. A size standard is indicated in kilodaltons. The uncropped images are presented in Supplementary Fig. 18. **c** Quantification of the PfCRT-specific signal strengths from 4 biologically independent samples (data points), normalized to the internal standard α-tubulin. A box plot analysis was overlaid over the individual data points, with the median (black line), mean (red lines), and 25% and 75% quartile ranges being shown. The whiskers above and below the box indicate the 90th and 10th percentile. The two datasets were not significantly different, according to the two-tailed Student's t-test. Source data are provided as a Source Data file[86].

cavity space (Fig. 3a). According to the model, the carboxyl side chain is solvent exposed and well placed to sense pH changes.

Several alternative mechanisms might explain the pH-sensing function of E207: (i) the ionization state of E207 affects binding of CQ in the sense that protons and CQ compete for binding to E207; (ii) E207 participates in the proton transfer across the pore; or iii) protonation and de-protonation of E207 induce conformational changes that affect the rate at which PfCRT transports substrates.

To test the first hypothesis, we computationally docked CQ into the substrate binding cavity of PfCRT[Dd2]. The docking process involved using a grid centered around E207, as well as protein configurations collected from three independent MD simulations for each PfCRT[Dd2] conformation and for each protonation state of E207 (see below). Each simulation was run for 200 ns and protein structures were collected every 10 ns from the 50 ns time point onwards, resulting in 45 configurations for each condition. Configurations were subsequently clustered and CQ was docked into the cluster centers (Fig. 3b, Supplementary Fig. 6 and Supplementary Table 2). In the most populated cluster of the open-to-vacuole conformation, CQ did not interact with E207 (Fig. 3b). In other clusters, CQ established non-specific hydrophobic interactions with E207 in some binding modes (Supplementary Fig. 6). The protonation state of E207 did not influence the outcome of the results. These findings suggest that the carboxyl group of E207 is not directly involved in initial CQ binding in the open-to-vacuole conformation, consistent with a previous report[36].

To explore if E207 plays a role in CQ binding during later transport steps, we modeled and simulated the occluded and the open-to-cytoplasm conformation, based on published models derived from the cryo-EM structure of PfCRT[7G8] and from the crystal structure of a triose-phosphate/phosphate translocator[27,37]. The latter approach has previously been used to model the open-to-vacuole and the occluded conformation of the wild type PfCRT variant from the CQ sensitive *P. falciparum* strain 3D7[38] (Supplementary Figs. 5 and 7). The same docking approach was followed as outlined above. Most binding modes of CQ were far from E207, or established non-specific hydrophobic interactions with E207 (Fig. 3b, Supplementary Fig. 6). Only in very few cases (4 out of 24 clusters) did CQ form a hydrogen bond with the carboxyl group of E207 - either in the protonated or deprotonated state or independent of the protonation status (Fig. 3b, Supplementary Fig. 6). These data show that the binding modes of CQ were mostly distant from E207 in the occluded and open-to-cytoplasm conformation, with only rare instances of hydrogen bonding with the carboxyl group of E207.

The goodness of the simulated protein conformations was assessed by monitoring the root mean squared deviation (RMSD) of the alpha-carbons in the whole protein and in the transmembrane helices (Supplementary Fig. 8), the root mean squared fluctuation (RMSF) of the alpha-carbons (Supplementary Fig. 9), and the stability of the secondary structure (Supplementary Fig. 8). The structures of the open-to-vacuole and occluded conformations were stable in the simulations, while the structure of the open-to-cytoplasm conformation was less stable, as evidenced by some loss of secondary structure during the simulation (Supplementary Fig. 8). This loss of secondary structure could be due to a lower quality of this model (Supplementary Table 3).

**Protons are noncompetitive inhibitors of CQ transport**

To further test the hypothesis that CQ and protons bind at distinct sites within the binding cavity of PfCRT, we performed an extended kinetic study and measured the CQ transport rates at the 60 min timepoint at 7 CQ concentrations (10 μM, 50 μM, 100 μM, 200 μM, 300 μM, 400 μM, 500 μM CQ) and four different pH values (4.5, 5.0, 5.5, and 6.0) (Fig. 4a). The transport rates were subsequently analyzed as a function of the CQ concentration and sixteen different models of inhibition were globally fit to the data, using the least-squares method. The plausibility of each model was subsequently evaluated on the basis of the Akaike information criterion difference ($\Delta AIC_c$) and the Akaike weight[39,40]. The most plausible model was partial noncompetitive inhibition (Fig. 4b). However, the mixed-type model and the two-site pure competitive model with $\Delta AIC_c$ values between 1 and 7 and

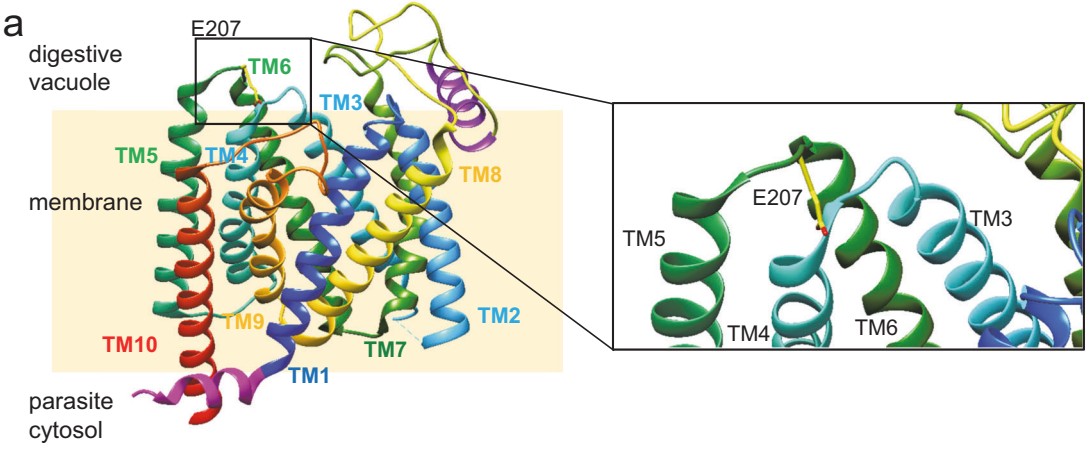

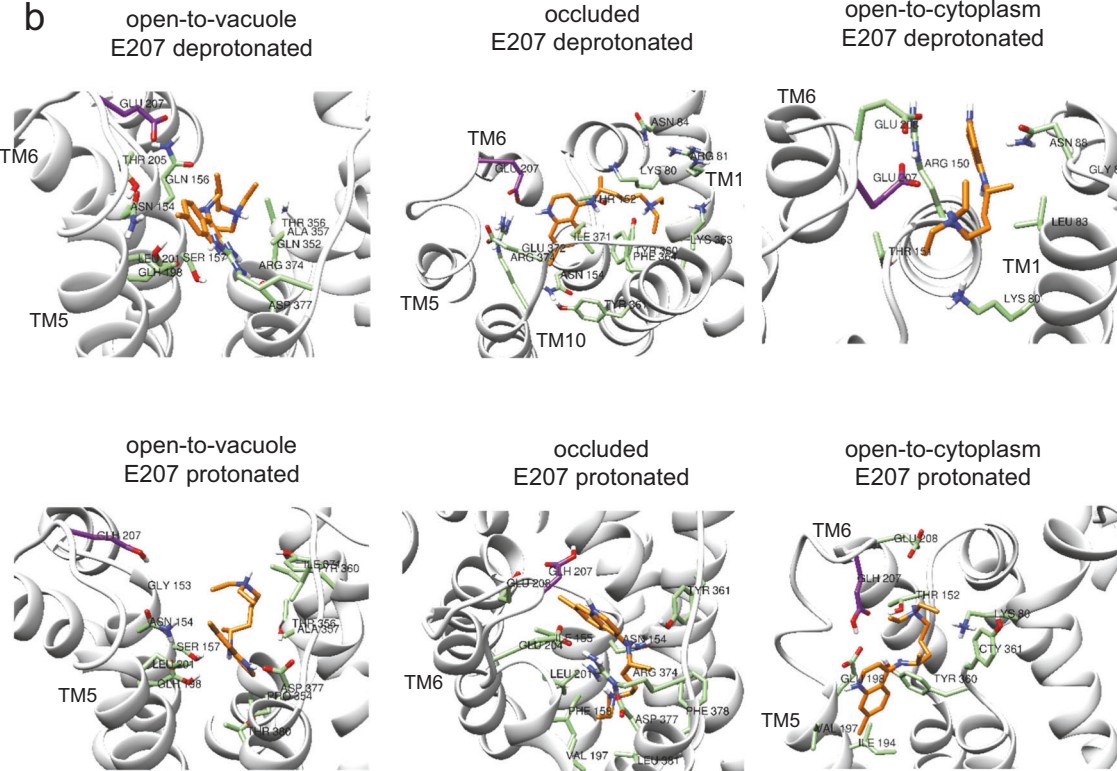

**Fig. 3 | I In silico docking of CQ to simulated conformations of PfCRT$^{Dd2}$.**
**a** Location of E207 (represented as sticks, with carbons in yellow) between transmembrane helices 5 and 6 (TM5 and TM6) in the open-to-vacuole conformation of PfCRT$^{Dd2}$. Note that the carboxyl side chain points into the cavity space. **b** Predicted binding modes of CQ (in orange) for different conformations of PfCRT (open-to-vacuole, occluded, open-to-cytoplasm) and different protonation states of E207 (in purple). Protein configurations were obtained from three independent 200-ns MD simulations, and clustered. CQ was docked to the centers of the clusters. Only the pose obtained for the most populated cluster is shown. Docking was performed using AutoDock Vina. Residues within a distance of 4 Å from CQ are shown in green. Source data are provided as a Source Data file[87].

Akaike weight > 0.02 could not be immediately dismissed, although they were less credible (Fig. 4b). We, therefore, re-examined the data using an $F$-test and compared the partial noncompetitive model with the two models ranked second and third. Again, partial noncompetitive inhibition provided the best fit to the kinetic data as compared with the partial mixed type ($F = 1.85$, $p = 0.19$, df = 23; $F$-test) and the two-site pure competitive model ($F = 0.025$, $p = 0.98$, df = 22; $F$-test).

The quality of the partial noncompetitive inhibition model was checked by superimposing the experimentally derived data points with the theoretical expectations in a 3D plot (Supplementary Fig. 10a). To assess the goodness of the fit, we determined the residuals (difference between the observed and expected uptake values) and plotted them as a function of the CQ and proton concentration (Supplementary Fig. 10b)[41]. The residuals were found to be randomly distributed across the x-axis within each CQ and proton concentration, indicating that the partial noncompetitive inhibition model can account for the interaction of PfCRT$^{Dd2}$ with CQ and protons. Moreover, a double reciprocal plot of the kinetic data revealed a family of straight lines with different slopes intersecting with the x-axis at the same point, again consistent with CQ and protons being partial noncompetitive inhibitors of one another on PfCRT$^{Dd2}$ (Fig. 4c).

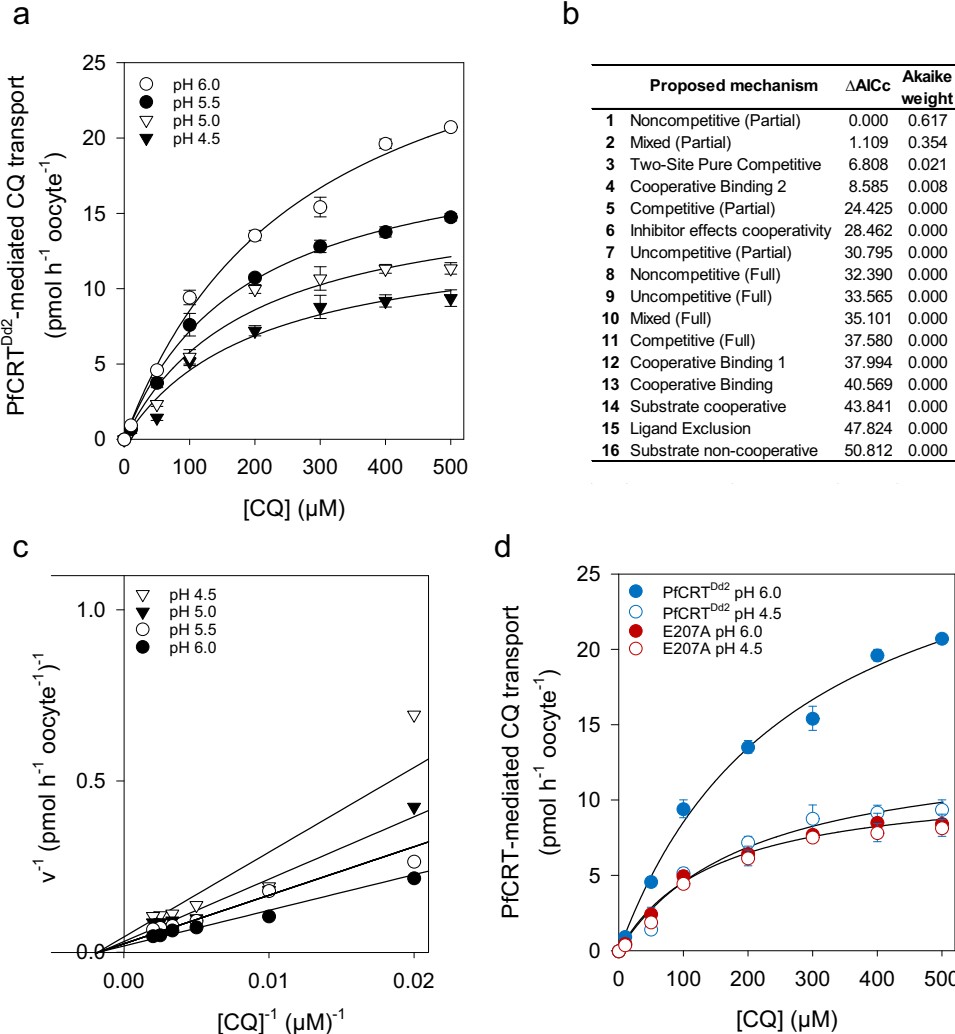

**Fig. 4 | I Effect of pH on PfCRT-mediated CQ transport kinetics. a** CQ transport kinetics of PfCRT[Dd2] at different proton concentrations (pH 4.5, 5.0, 5.5, and 6.0). Each data point represents the mean ± SEM of at least 3 biologically independent samples (range 3 to 14). **b** Sixteen different models of inhibition were fitted to the data represented in (**a**) using the least-squares methods. Plausibility was analyzed by calculating the Akaike information criterion difference ($\Delta AIC_c$) and the Akaike weight. Models are shown in descending order. The kinetic parameters derived from the two most plausible inhibition models are presented in Table 1. **c** Double reciprocal Lineweaver-Burk Plot obtained with the data described in (**a**). **d** CQ transport kinetics of PfCRT[Dd2] and E207A at a pH of 4.5 and 6.0. Note, the E207A variant exhibits comparable CQ transport kinetics at pH 4.5 and 6.0. Each data point represents the mean ± SEM of at least 3 biologically independent samples (range 3 to 6). See Table 2 for kinetic parameters. Source data are provided as a Source Data file[86].

Partial noncompetitive inhibition describes a kinetic model in which the transporter or enzyme can bind the substrate and the inhibitor at independent sites, allowing for the simultaneous binding of both molecules. The affinity of the inhibitor to the transporter is unaffected by pre-bound substrate, but binding of the inhibitor reduces the transport activity, albeit not to zero, even at infinite inhibitor concentrations. Taken together, the docking and the kinetic analyses support an allosteric binding model in which PfCRT[Dd2] can simultaneously accept both protons and CQ with unchanging affinities and in which increasing proton concentrations reduce the CQ transport rate to a baseline value. The kinetic parameters obtained for the partial noncompetitive binding model are listed in Table 1 and show a Michaelis Menten constant, $K_M$, of 220 ± 25 μM for CQ, consistent with previous reports[22,34], a $K_i$ of 1.8 ± 0.6 μM for protons and a factor β of 0.36 ± 0.02, the latter indicating the change in the CQ transport rate when a proton is already bound to the transporter. Unlike PfCRT[Dd2], E207A displayed pH-insensitive CQ transport kinetics, with statistically indistinguishable CQ concentration curves at pH 4.5 and 6.0 ($F = 2.26$, $p = 0.15$, df=14) (Fig. 4d and Table 2).

## E207 is not involved in the proton transfer pathway

In case of an involvement of E207 in proton coupling, an extended proton transfer pathway would be required to span the entire channel from the peripheral position of E207 at the digestive vacuolar site to the cytoplasmic site. Although several putative hydrogen bond networks were identified in static models of the three conformations (open-to-vacuole, occluded, and open-to-cytoplasm state), using the program Cgraphs[42], none involving E207 span the membrane (Supplementary Fig. 11). Furthermore, some of the E207 networks include interactions with the backbone, which would be unusual for a proton transfer pathway as this typically relies on titratable residues[43,44]. To identify more plausible candidates for proton transfer, we re-examined the alanine mutants, but this time with a different set of criteria in mind. As shown in other systems, residues involved in proton-binding and proton-transfer are titratable and frequently consist of Glu and Asp; they are present in a transmembrane domain and are located in the center of the channel; they are able to engage in inter-helical H-bonding, frequently with Ser or Thr, for coupling protonation dynamics with conformational changes; and replacing them strongly

affects the transport activity[45–47]. Applying these criteria identified D137 and D329 as initial candidates. Both residues are located in transmembrane domains (TM3 and TM8, respectively; Supplementary Fig. 2) and replacing them by the isosteric amino acid Asn drastically reduced the transport activity for CQ at pH 6.0 (Fig. 1d). Both residues are conserved among *Plasmodium* spp (Supplementary Fig. 4a) and structural models placed both residues in the center of the channel (Supplementary Fig. 12). MD simulations revealed that D137 forms an inter-helical hydrogen bond with S227 (residing on TM 6; distance of 1.85 Å) in the open-to-vacuole and the occluded conformation (Supplementary Figs. 12a, b, and 13), consistent with a previous report[36]. In comparison, D329 engages in intra-helical hydrogen bonding - with T333 (like D329 also residing on TM 8; distance of 2.26 Å) (Supplementary Figs. 12c, d, and 13). These initial findings would favor D137 as a plausible candidate for proton shuttling, although further experiments are needed to clarify the role of D137 and also that of D329 in the transport cycle.

## E207 is involved in the transport cycle

E207 appears quite isolated in the modeled open-to-vacuole conformation of PfCRT[Dd2]. Interestingly, the surroundings of E207 drastically changed in subsequent transport states. In the occluded conformation, E207 is in close proximity to residues contained within TM1, according to MD simulations. In particular, the E207 carboxyl side chain interacted via a salt bridge with the amino group of K80 (distance, ~1.70 Å) and via a hydrogen bond with the carboxamide group of N84 (distance, ~1.85 Å) (Fig. 5a, b; Supplementary Figs 14, 15). However, these interactions only occurred when E207 was in its carboxylate form, i.e., deprotonated and charged. In the protonated form, E207 formed a

hydrogen bond with E372 located in the loop between transmembrane domain 9 and 10 (distance 1.8 Å) (Fig. 5c, d; Supplementary Figs 14, 15). Similarly, MD simulations predicted E207 to interact with K80 and N84 exclusively in its charged, deprotonated form in the open-to-cytoplasm conformation, although the goodness of the prediction was low due to the poorer quality of the simulations and the modeled protein structure (Supplementary Figs 14, 15). A carboxylate side chain involved in a salt bridge is expected to have a lower pKa value since the salt bridge stabilizes the negative charge[48]. In the case of the carboxylate side chain of E207, the estimated pKa value is indeed significantly lower in the occluded conformation than in the open-to-vacuole conformation ($3.8 \pm 0.1$ and $4.2 \pm 0.1$, respectively; $n = 45$; $p = 0.01$ according to $t$-test) (Supplementary Table 4). Conversely, the protonated E207 has a significantly increased estimated pKa value of $5.3 \pm 0.08$ in the occluded state, compared with $4.4 \pm 0.06$ in the open-to-vacuole conformation ($n = 45$; $p < 0.001$; according to $t$-test) (Supplementary Table 4). In the simulations of the open-to-vacuole conformation, E207 remained in a peripheral, isolated position and there was no evidence for an interaction with K80, N84, or E372 (Supplementary Figs 14, 15), consistent with a previous report[36]. These findings suggest that E207 approaches K80, N84, and E372 during the transport cycle and that the preference for interactions with either K80 and N84 or E372 depends on the protonation state of the carboxyl side chain.

To test for these predicted interactions of E207 in the occluded and open-to-cytoplasm conformation, we replaced the putative interaction partners by alanine and determined the transport activities at pH 6.0 and 4.5. As seen in Fig. 6a, the K80A mutation displayed an almost pH independent CQ transport activity, with an $R$-value of $0.76 \pm 0.04$, while maintaining a transport activity at pH 6.0 comparable to that of E207A. In comparison, the N84A and the E372A mutants were still largely pH sensitive, although their $R$-values were significantly higher than that of PfCRT[Dd2] (0.42 and 0.49, respectively; $p < 0.001$, Holm-Sidak ANOVA). These findings suggest a critical interaction between E207 and K80 during transition from the open-to-vacuole to the occluded conformation. To investigate this conclusion further, we generated a double mutant in which we swapped the two residues. The resulting E207K/K80E double mutant was pH sensitive ($R$-value of $0.54 \pm 0.18$) and, importantly, had a transport activity at pH 6.0 comparable to that of PfCRT[Dd2] ($4.75 \pm 0.91$ pmol h$^{-1}$ oocyte$^{-1}$) (Fig. 6a).

## Functional substitution of Asp and His for E207

To test the hypothesis of position 207 playing a critical role as a hydrogen acceptor, we replaced E207 with any of the other 19 proteinogenic amino acids (including the Ala replacement). Most of the 19 variants generated had high $R$-values, indicative of carriers with limited pH sensitivity (Fig. 6b and Supplementary Table 5). This included the positively charged amino acids Arg and Lys (side chain pK$_a$ values of 12.48 and 10.79[30], respectively). Only Asp and His could functionally substitute for E207, at least to a large extent, albeit their $R$-values were slightly higher than that of PfCRT[Dd2] in the following order: H > D > E

## Table 1 | Kinetic parameters describing interactions of PfCRT[Dd2] with CQ and protons

| parameter | kinetic model | | | |
|---|---|---|---|---|
| | noncompetitive (partial) | | mixed (partial) | |
| | mean ± SEM | 95% CI | mean ± SEM | 95% CI |
| $V_{max}$ (pmol h$^{-1}$ oocyte$^{-1}$) | 40 ± 4 | 30.53–44.51 | 45 ± 9 | 28.1–61.75 |
| $K_S^{CQ}$ (µM) | 220 ± 25 | 170–268 | 320 ± 100 | 123–517 |
| $K_S^{H+}$ (µM) | 1.80 ± 0.60 | 0.61–2.98 | 2.54 ± 1.08 | 0.43–4.66 |
| α | | | 0.53 ± 0.26 | 0.02–1.04 |
| β | 0.36 ± 0.02 | 0.31–0.40 | 0.27 ± 0.06 | 0.15–0.40 |

$V_{max}$, the maximum velocity of substrate transport, $K_S^{CQ}$ and $K_S^{H+}$, the dissociation constants for CQ-PfCRT[Dd2] and H$^+$-PfCRT[Dd2] complexes, respectively; α, the factor by which these $K_S$ values change when the other substrate is already bound to the transporter; β, the factor by which the $V_{max}$ is affected by the inhibitor; CQ, chloroquine, H$^+$, protons. CI, confidence interval. Shown are the best-fit values of kinetic parameters obtained using the partial noncompetitive and partial mixed-type inhibition models.

## Table 2 | Kinetic parameters describing PfCRT[Dd2]-, E207A-, and E207H-mediated CQ transport at pH 4.5 and pH 6.0

| PfCRT variant | | $V_{max} \pm$ SEM pmol h$^{-1}$ oocyte$^{-1}$ | $K_M \pm$ SEM µM | $p$ value |
|---|---|---|---|---|
| PfCRT[Dd2] | pH 6.0 | 32 ± 2 | 270 ± 50 | $1.1 \times 10^{-10}$ (df = 12, $F$ = 182.27) |
| PfCRT[Dd2] | pH 4.5 | 14 ± 2 | 200 ± 60 | |
| E207A | pH 6.0 | 11 ± 1 | 150 ± 30 | 0.15 (df = 12, $F$ = 2.26) |
| E207A | pH 4.5 | 11 ± 1 | 180 ± 30 | |
| E207H | pH 6.0 | 29 ± 1 | 230 ± 20 | $1.4 \times 10^{-5}$ (df = 12, $F$ = 32.72) |
| E207H | pH 4.5 | 13 ± 1 | 230 ± 35 | |

Statistical significance was determined between the CQ transport activity at pH 4.5 and 6.0, using a two-tailed $F$-test.
$V_{max}$ maximal CQ transport velocity, $K_M$ Michaelis–Menten constant, $SEM$ standard error of the mean, $df$ degrees of freedom, $F$ F-factor.

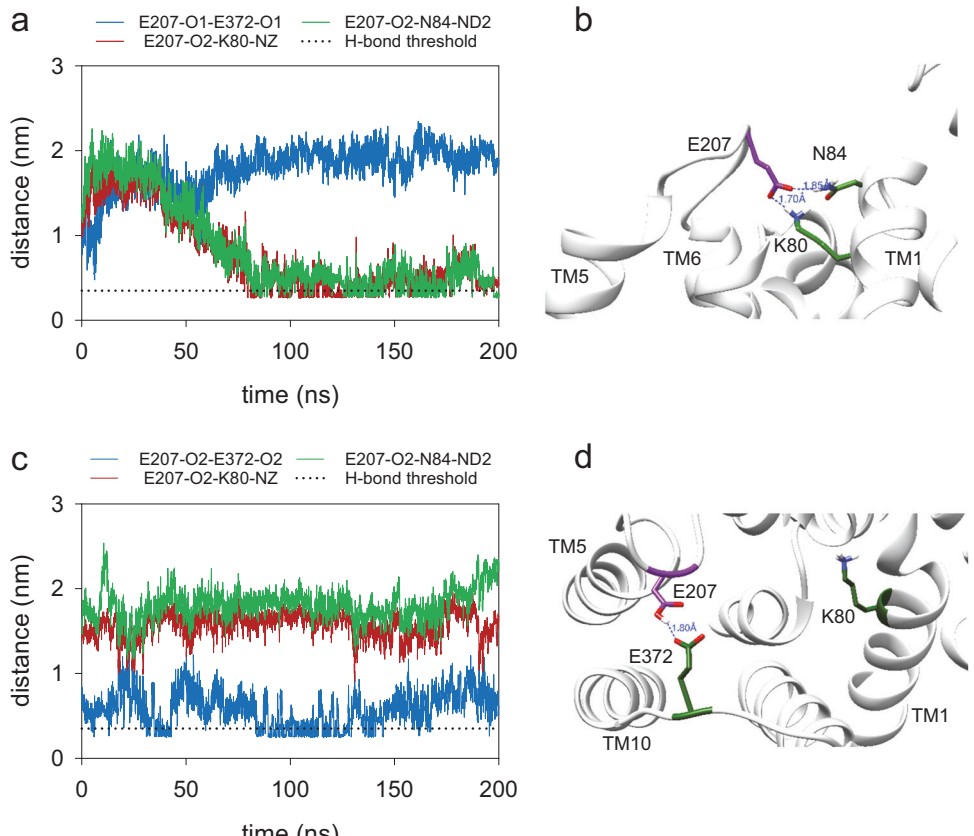

**Fig. 5 | I Effect of the protonation state of E207 on structural changes in the occluded conformation of PfCRT$^{Dd2}$. a** Distance between hydrogen bond donors and acceptors, E207 and E372/K80/N84, in one MD simulation of the occluded conformation of PfCRT$^{Dd2}$, with E207 deprotonated. Dotted line indicates the threshold distance of 3.5 Å for the formation of a hydrogen bond. **b** Snapshot from one MD simulation illustrating the hydrogen bonds between E207 and K80 and N84 in the occluded conformation, with E207 deprotonated. **c** Distance between hydrogen bond donors and acceptors, E207 and E372/K80/N84, in one MD simulation of the occluded conformation of PfCRT, with E207 protonated. **d** Snapshot from one MD simulation illustrating hydrogen bond formation between E207 and E372 in the occluded conformation, with E207 protonated. Source data are provided as a Source Data file[86].

(Fig. 6b). For the His-containing variant, we measured the CQ transport activity over the pH range from 4.5 to 7.0, yielding a pH optimum curve comparable to that of PfCRT$^{Dd2}$ (Fig. 6c). Furthermore, a full kinetic analysis of the E207H mutant revealed $K_M$ and $V_{max}$ values comparable to those of PfCRT$^{Dd2}$ both at pH 4.5 ($F = 0.26$, $p = 0.78$, df=12; $F$-test) and pH 6.0 ($F = 1.53$, $p = 0.26$, df = 12; $F$-test) (Fig. 6d and Table 2).

The finding that Asp could replace Glu was expected because of similar physico-chemical properties. Both amino acids have comparable side chain p$K_a$ values of 3.65 and 4.25, respectively[30], and they change their charge from negative to neutral upon protonation. Having a p$K_a$ of 6.0, His can also be protonated and deprotonated within the operative range of PfCRT and it can also serve as a hydrogen bond acceptor. However, unlike Glu and Asp, His becomes positively charged upon protonation. Although MD simulations of the E207H mutant did not detect hydrogen bond formation between the His imidazole side chain and K80 or any other residue, this may be due to the lack of electrostatic attraction and, consequently, of a weaker interaction. Nonetheless, these findings deemphasize a possible role charge might play in the function of E207. Instead, the findings further support the hypothesis that position 207 requires a titratable residue that can accept a hydrogen for bond formation during conformational transitions of the carrier.

## Discussion

Many transporters are pH regulated and PfCRT$^{Dd2}$ is no exception. What sets PfCRT$^{Dd2}$ apart is the fact that it operates 40% below its maximal CQ transport activity at the pH of ~5.2 of the digestive vacuole (Fig. 1a). Usually, transporters are well adapted to their environment and show maximal transport activity around the pH of their location. Why the strong pressure exerted by drugs on PfCRT was unable to shift the activity optimum is unclear as is the structural basis of the pH dependence. Here, we have investigated the structural and functional basis of the pH sensitivity displayed by PfCRT$^{Dd2}$. Our study identified a single amino acid, E207, as a critical pH sensor and suggests that the pH dependence is an intrinsic feature of the transport process itself, with the ionized carboxyl group of E207 acting as a hydrogen acceptor for interactions that are predicted to accelerate progression through the transport cycle.

E207 was identified in a systematic screen that involved residues titratable within the operative pH range of PfCRT$^{Dd2}$ and aromatic amino acids capable of cation–π interactions (Fig. 1d). The pH-insensitivity of the E207A variant was accompanied by a reduced overall transport activity. This is not unusual and has also been reported for other pH-dependent transporters, including the bacterial arginine-agmatine antiporter[31] and the human nucleoside transporter 3[32]. The phenomenon is explained in other systems by pH-sensing residues also contributing to the overall tertiary structure of the carrier, in addition to their pH-sensing function. In the case of E207, we do not think that an unspecific structural effect can fully account for the decline in transport activity observed for most of the E207 replacement mutants. Our data are more in line with E207 having a defined role in the transport cycle and that the pH sensing activity of E207 is a by-product of this function.

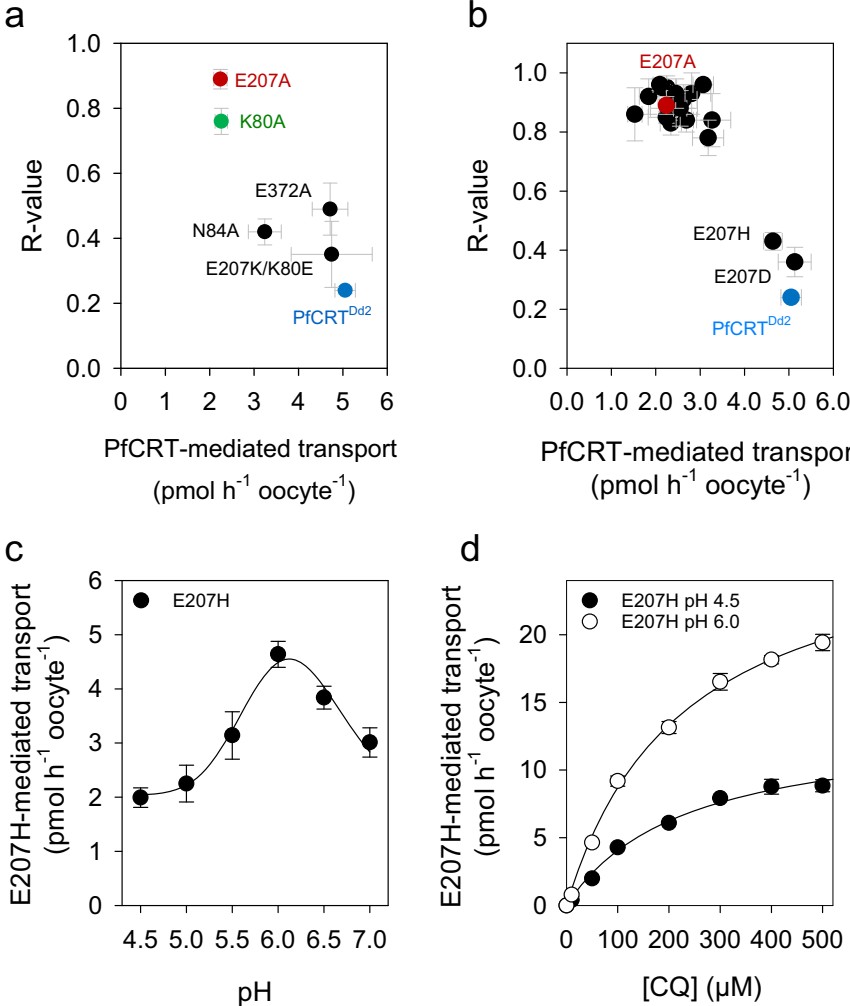

**Fig. 6 | Rescue of pH-sensitive CQ transport. a** Alanine mutagenesis of potential E207 interaction partners in the occluded conformation. Mutants were generated in the PfCRT[Dd2] background. Shown is a plot of *R*-values against the CQ transport activity at pH of 6.0 of the PfCRT variants indicated. Specific CQ transport was determined from an external CQ concentration of 50 µM. Each data point represents the mean ± SEM of n biologically independent samples. The number of biologically independent samples per mutant are: 14, PfCRT[Dd2]; 9, E207A; 4, E327A; 4, K80A, 4, N84A; 3, E207K/K80E. **b** Replacement mutagenesis of E207. E207 was replaced by each of the 19 proteinogenic amino acids in the PfCRT[Dd2] background and the *R*-value was determined as a function of the CQ transport activity at a pH of 6.0. Note, only Asp and His (E207D and E207H variant) were able to reconstitute pH-sensitive CQ transport. Each data point represents the mean ± SEM of at least 3 biologically independent samples (range 3 to 14). **c** Effect of pH on E207H-mediated CQ transport. Each data point represents the mean ± SEM of 3 biologically independent samples. **d** E207H-mediated CQ transport kinetics at a pH of 4.5 and 6.0. Each data point represents the mean ± SEM of at least 3 to 4 biologically independent samples. Kinetic parameters are listed in Table 2. Source data are provided as a Source Data file[86].

If the pH-sensing ability of E207 does serve a physiological purpose, then it might prevent detrimental uncoupled proton leakage via PfCRT in case of an acid overload in the parasite's digestive vacuole, as shown in other systems[49].

E207 is conserved among most *Plasmodium* species, with the exception of the four rodent malaria parasites, *P. berghei*, *P. chabaudi*, *P. vinckei* and *P. yoelii* (Supplementary Fig. 4). The evolutionary distance between PfCRT and its orthologues in rodent malaria parasites might result from adaptive processes that have led to a differentiation of the physiological functions exerted by CRT and the mechanisms regulating CRT activity. Unfortunately, very little is known about the natural substrate of CRT in other *Plasmodium* species. There is, however, evidence that the mode of action of CQ and the mechanism of resistance varies among Plasmodial species and involves factors other than CRT, such as the digestive vacuolar amino acid transporter AA1[50,51].

To assess the function of E207 in the context of PfCRT[Dd2], we relied on structural models generated based on the published cryo-EM structure of the related isoform, PfCRT[7G8], showing the transporter in the open-to-vacuole conformation in association with a phage display antibody[27], and on protocols previously established to model the occluded state of the wild type PfCRT and the open-to-cytoplasm conformation of PfCRT[7G8] [27,38]. We acknowledge that our models are only predictions of the tertiary structure of PfCRT[Dd2] and that further structural information at a resolution of <1.8 Å[52] would be required to verify some of the hypotheses formulated in this study on the role E207 plays in the transport cycle. Such high-resolution maps are currently not available and even the published cryo-EM structure has only a resolution of 3.2 Å[27], which is far below the <1.8 Å resolution necessary to demonstrate hydrogen bond formation and salt bridges in static structures[52]. It should further be evaluated which role E207 plays during the transport of oligopeptides, the natural substrate of PfCRT. Regardless, we do not think that these limitations devalue our findings or the conclusions drawn from them.

E207 resides at the entrance of the substrate binding cavity in the open-to-vacuole conformation of the carrier, with its carboxyl side

chain being exposed to the digestive vacuolar environment. Such an exposed position predestines E207 as a pH sensor, but does not necessarily exclude other effector mechanisms, such as a role in CQ binding, proton transfer or the transport cycle. A role of E207 in CQ binding appears unlikely, based on docking and MD simulations. The binding modes of CQ were mostly distant from E207 and only in 10% of clusters examined was there a specific interaction via a hydrogen bond between the E207 carboxyl side chain and CQ (Fig. 3b and Supplementary Fig. 6). Further support for E207 playing no role in CQ binding came from kinetic studies of CQ transport at different pH values, which showed that protons and CQ are partial noncompetitive inhibitors of one another and, thus, bind at separate sites (Fig. 4). Cases of noncompetitive inhibition are frequently observed for pH- and inorganic ion-dependent enzymatic and transport reactions[53].

We also found no evidence of E207 partaking in a putative proton transfer. The peripheral position of E207 would require an elaborated pathway to cross the pore. Although E207 is involved in hydrogen bond networks, these hydrogen bond networks involve the immediate surroundings of E207 and do no extend all the way from one site of the transporter to the other. Furthermore, some of the E207 hydrogen bond networks also included interactions with the protein backbone, which would be unusual for a proton transfer pathway (Supplementary Fig. 11).

A more promising candidate for proton binding and transfer is residue D137 on TM 3. Inserting a negatively charged residue in a transmembrane domain is energetically unfavorable ($\sim$4–5 kcal mol$^{-1}$) and is, therefore, avoided unless the residue plays a functional role, e.g., in proton transfer[46]. To couple protonation dynamics with conformational changes, negatively charged residues frequently engage in inter-helical hydrogen bond formation via their carboxylate group with the hydroxyl group of Ser or Thr[46]. This particular inter-helical carboxyl–hydroxyl interaction appears to be a hallmark of a proton-transfer mechanism in a membrane transporter[46]. MD simulations indicated that D137 forms inter-helical hydrogen bonds with S227 on TM 6 in both the open-to-vacuole and occluded conformations, supporting its involvement in proton transfer (Supplementary Fig. 12). Additionally, substitution of D137 with the isosteric amino acid Asn greatly reduced CQ transport activity, consistent with the role of D137 in proton shuttling (Fig. 1d). While initially considering D329 as a potential candidate for proton transfer, we favored an alternative function due to its absence of inter-helical hydrogen bonding with Ser or Thr, such as an involvement in substrate binding[36]. Further experiments, including constant pH MD simulations, are necessary to fully elucidate the functions of both residues in the transport cycle of PfCRT$^{Dd2}$.

While E207 appears isolated in the open-to-vacuole conformation, its surroundings are predicted to drastically change during the transport cycle. In the MD simulation of the occluded conformation, E207 no longer resides in a peripheral location, but instead relocated towards K80 and N84 on transmembrane domain 1 (Fig. 5a and b). The distance between the carboxylate side chain of E207 and the amino group of K80 of 1.7 Å is in a range where a salt bridge can be formed between them, with the negatively charged carboxyl group of E207 drawing a proton from the positively charged side chain amino group of K80 (Fig. 7) (pK$_a$ of amino side chain of Lys, 10.53). pK$_a$ estimations corroborate such a model, suggesting that the pK$_a$ of the E207 carboxylate side chain decreases from $4.2 \pm 0.1$ in the open-to-vacuole conformation to $3.8 \pm 0.1$ in the occluded conformation ($n = 45$; $p = 0.01$; $t$-test) (Supplementary Table 4). A drop in the pKa by 0.4 units is in the range of 0.2 to 4.0 units reported for carboxylate side chains involved in salt bridges[48]. Moreover, replacing K80 by Ala resulted in a transport phenotype similar to that of E207A, i.e., a comparable, almost pH-independent transport activity (Fig. 6a). The importance of the interaction between E207 and K80 is further supported by the E207K/K80E charge-swap variant that restored pH sensitivity and CQ

transport activity at pH 6.0 (Fig. 6a). A functional role of E207 as a proton acceptor is also consistent with our replacement mutagenesis, showing that only His and Asp can substitute for E207 but none of the other proteinogenic amino acids (Fig. 6b). Both His and Asp are titratable residues that are capable of forming a hydrogen bond with K80. Electrostatic interactions might further strengthen the interaction with K80, at least in the case of Glu and Asp. Bond-strengthening electrostatic interactions would not occur between His and K80, which might explain why the E207H mutant had an $R$-value significantly higher than those of PfCRT$^{Dd2}$ and E207D (Fig. 6b). When the carboxyl group of E207 is protonated, E207 also relocated in the occluded state, but in this case, it moved towards E372, according to MD simulations (Fig. 5c, d). This interaction would involve hydrogen bond formation with the carboxyl side chain of E372 (Fig. 5c, d). Substituting Ala for Glu at position 372 had a partial effect on the $R$-value, with the mutant still being pH dependent, albeit less so than PfCRT$^{Dd2}$ (Fig. 6a).

On the basis of our findings, we postulate that E207 plays a critical role in the drug-transporting function of PfCRT$^{Dd2}$ by accelerating progression through the transport cycle via salt bridge formation with K80 as the carrier transits from the open-to-vacuole conformation to the occluded state (Fig. 7). In comparison, the E207A variant is unable to interact with K80 and, thus, represents a pH-insensitive but poorer version of PfCRT$^{Dd2}$. Its residual activity may be driven by proton-cotransport[25,26] or be induced by CQ binding. Similarly, if the carboxyl side chain of E207 is protonated, E207 would no longer be able to serve as a proton acceptor, which, in turn, would prevent it from interacting with K80, resulting in a reduced transport activity (Fig. 7). Furthermore, the protonated E207 would not relocate towards K80 but instead towards E372 (Fig. 7), with further implications for the transport rate. The additional proton may also cause spatial problems during the transport process. It is, therefore, advantageous for the parasite to preserve the pH sensitivity of PfCRT$^{Dd2}$ and operate below maximal turn-over rates at the pH of ~5.2 of the digestive vacuole since mutating E207 would yield even less transport activity.

## Methods
### Ethics approval
Ethical approval of the work performed with the *Xenopus laevis* frogs was obtained from the Regierungspräsidium Karlsruhe (Aktenzeichen 35-9185.81/G-31/11 and 35-918581/G-21/23) in accordance with the German "Tierschutzgesetz."

### Reagents and radiolabelled compounds
[³H]-CQ (specific activity, 37 Ci mmol$^{-1}$), [³H]-quinine (specific activity, 20 Ci mmol$^{-1}$), [³H]-quinidine (specific activity, 20 Ci mmol$^{-1}$), [³H]-vinblastine (specific activity, 7.8 Cimmol$^{-1}$), [³H]-vincristine (specific activity, 5 Cimmol$^{-1}$), [³H]-amantadine (specific activity, 33 Cimmol$^{-1}$) and [³H]-piperaquine (specific activity, 15 Cimmol$^{-1}$) were obtained from American Radiolabeled Chemicals, GE Healthcare or Quotient Biorsearch. Other chemicals were provided by Sigma-Aldrich.

### Site-directed mutagenesis of PfCRT
Site-directed mutagenesis of PfCRT$^{Dd2}$ was performed as described using the megaprimer method with overlap extension[21,54]. The primer pairs used are listed in Supplementary Table 6. Please note that a codon-optimized and oocyte-adapted version of PfCRT$^{Dd2}$ was used[22] (Supplementary Fig. 16). The resulting DNA fragment was cloned into the SP64T expression vector and verified by sequencing[22].

### PfCRT expression in *X. laevis* oocytes
Oocytes were obtained as described previously[24]. Briefly, adult (>2 years old) female *X. laevis* frogs (NASCO) were anesthetized in a cooled solution of ethyl 3-amino benzoate methanesulfonate (0.1%, w/v), followed by a surgical removal of parts of the ovary. The ovary lobes were carefully dissected and small pieces of oocytes were incubated in

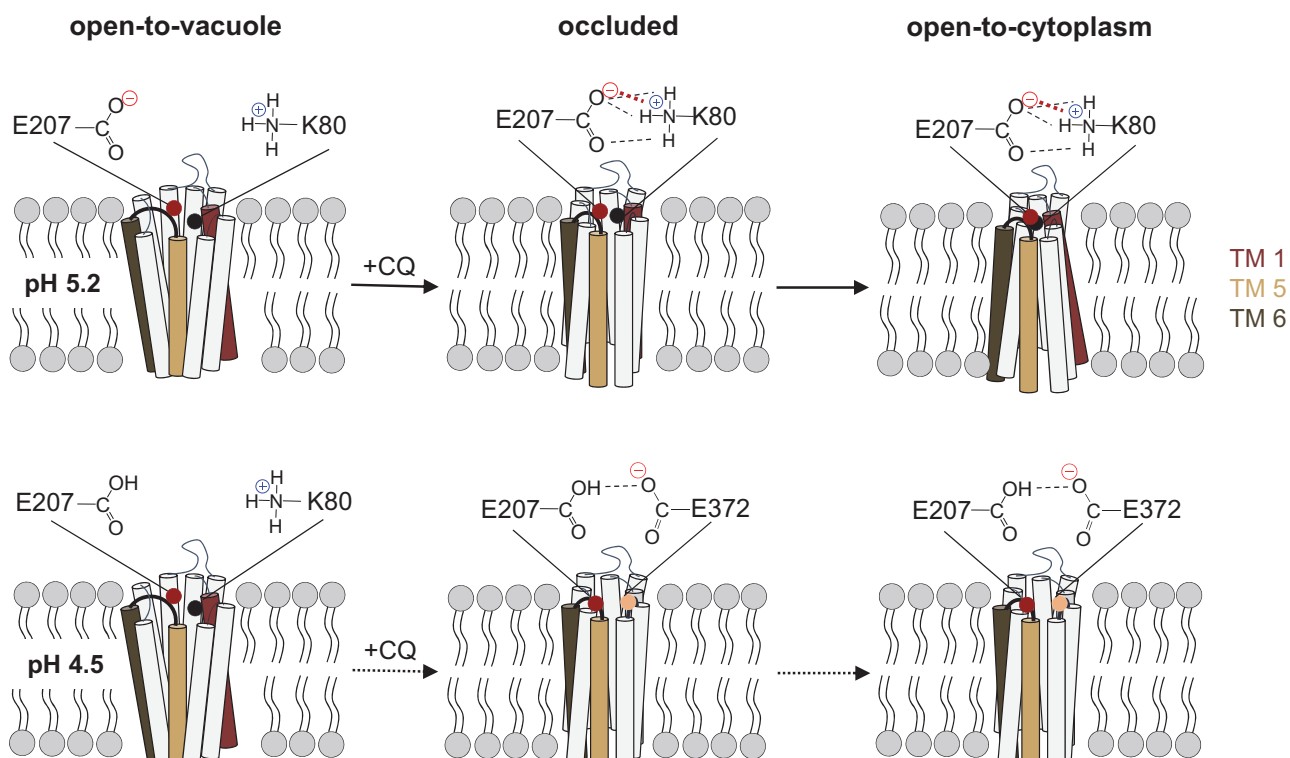

**Fig. 7 | Model of the pH-sensing mechanism of PfCRT^Dd2.** *Upper panel.* The carboxyl side chain of E207 is located at the periphery of the substrate binding cavity in the open-to-vacuole conformation, with the side chain protruding into the cavity space and, thus, being exposed to the environment. At the pH of 5.2 of the digestive vacuole, the carboxyl side chain is mostly deprotonated and, hence, negatively charged. However, some protonated species exist, which explains the reduced transport rate at pH 5.2 as compared with pH 6.2. During the transport cycle, E207 repositions from a peripheral to an engaged position, moving the negatively charged carboxyl side chain close to the positively charged amino group side chain of K80. The predicted close distance between these two residues of ~ 1.80 Å facilitates the formation of a salt bridge, which, in turn, accelerates progression through the transport cycle. *Lower panel.* Protonation of the carboxyl side chain of E207 at low pH values would preclude the formation of a salt bridge with K80. Instead, the protonated carboxyl side chain of E207 would hydrogen bond with the carboxyl group of E372. In addition, the additional proton might cause possible steric problems. As a result, the transport activity is reduced.

Ca$^{2+}$-free ND96 buffer (96 mM NaCl, 2 mM KCl, 1 mM MgCl$_2$ buffered with 5 mM HEPES/NaOH, pH 7.5) supplemented with collagenase D (0.1%, w/v; Roche), BSA (0.5%, w/v), and 9 mm Na$_2$HPO$_4$ at 18 °C for 14-16 h while gentle shaking. After incubation, the oocytes were carefully washed several times with ND96 buffer (96 mM NaCl, 2 mM KCl, 1.8 mM CaCl$_2$, 1 mM MgCl$_2$ buffered with 5 mM HEPES/NaOH, pH 7.5) supplemented with 100 mg l$^{-1}$ gentamycin). Defolliculated and healthy-looking stage V-VI oocytes were manually selected for RNA microinjection. The RNA was generated as following: The codon-optimized sequences of PfCRT variants were cloned into the pSP64T expression vector. After vector linearization using the restriction endonucleases BamHI or SalI (New England Biolabs), transcription was performed using the in vitro SP6 mMessage mMachine Kit (Ambion)[34]. The obtained RNA was kept at −80 °C and diluted with nuclease-free water to a concentration of 0.6 µg µl$^{-1}$. Precision-bore glass capillary tubes (3.5-inch glass capillaries; Drummond Scientific Co.) were pulled on a vertical puller (P-87 Flaming/Brown micropipette puller, Sutter Instrument Co.), graduated, and placed on a microinjector (Nanoject II Auto-Nanoliter Injector, Drummond Scientific Co.). 50 nl of nuclease-free water alone (control oocytes) or 50 nl of 0.6 µg µl$^{-1}$ RNA were injected under stereomicroscopic control. Injected oocytes were incubated for 46–72 h at 18 °C with twice-daily buffer changes before usage in experiments.

**CQ uptake assay**
Drug uptake assays were performed as described previously[21,22,24,34]. Briefly, oocytes were incubated for 1 h at room temperature in ND96 buffer (pH 4.5 and 6.0) supplemented with 50 µM unlabeled drug and [$^3$H]CQ (50 nM), [$^3$H]quinine (62.5 nM), [$^3$H]quinidine (62.5 nM), or [$^3$H]

piperaquine (40 nM). The direction of radio-labeled drug transport is here from the mostly acidic extracellular solution (pH 4.5 or 6.0, as indicated) into the oocyte cytosol (pH ~ 7.2), which corresponds to the efflux of drug from the acidic digestive vacuole (pH 5.2) into the parasite cytoplasm (pH 7.2). The topology of PfCRT in the oolemma is such that both the N- and C-terminus face the extracellular milieu[3,22]. In the parasite, the N- and C-terminus face the lumen of the digestive vacuole[23]. CQ uptake kinetics were performed by incubating oocytes in ND96 buffer adjusted to a pH of 4.5, 5.0, 5.5, or 6.0 and supplemented with 50 nM [$^3$H]CQ and one of 7 different CQ concentrations (10 µM, 50 µM, 100 µM, 200 µM, 300 µM, 400 µM, 500 µM CQ). The reaction was stopped at the 60 min time point, which is in the linear uptake range[22,34] (Supplementary Fig. 17). For determination of the amount of drug uptake for each oocyte, the reaction medium was removed and the oocytes were washed three times with 2 ml of ice-cold ND96 buffer. Each oocyte was individually transferred to a scintillation vial containing 5% sodium dodecyl sulfate (200 µl) for lysis. The radioactivity of each sample was measured using a liquid scintillation analyzer Tri-Carb 4910 TR (PerkinElmer). Oocytes that were injected with water instead of RNA were analyzed in parallel, representing the uptake of a drug by diffusion and/or endogenous permeation pathways. The resulting value represents the "background" level of drug accumulation in oocytes (in-depth discussion in[22]). The specific PfCRT-mediated uptake was determined by subtracting the corresponding background value measured in water-injected oocytes from that of PfCRT-expressing oocytes. In all cases, at least three separate experiments were performed on oocytes from different frogs (biological replicate), and for each condition in an experiment, measurements were made

from 10 oocytes per treatment. For pH 5.5, 6.0, and 6.5, ND96 was buffered with 5 mM MES/Tris base. For pH 4.5, 5 mM Homo-Pipes was used as a buffer and 5 mM HEPES for uptake at pH 7.0.

### Western blot analysis of oocyte lysates

Western blot analysis of oocytes was performed as described previously[55]. Briefly, 3 days after RNA injection, total lysates were prepared from *X. laevis* oocytes by adding 20 µl of radio-immunoprecipitation assay-lysis buffer (10 mM HEPES-Na, pH 7.4, 150 mM NaCl, 1 mM EDTA, 1% Nonidet P-40, 0.5% sodium deoxycholate, 0.1% SDS) supplemented with protease inhibitors (Complete, Roche Applied Science). After removing of cellular debris by centrifugation, lysates were mixed with two volumes of sample buffer (250 mM Tris, pH 6.8, 3% SDS, 20% glycerol, 0.1% bromophenol blue). Then, the extracts were size-fractionated using 12% SDS-PAGE and transferred to a polyvinylidene difluoride membrane. The following antibodies were used: guinea pig anti-PfCRT antiserum (raised against the N terminus of PfCRT (MKFASKKNNQKNSSK); 1:1000 dilution; Eurogentec), donkey anti-guinea pig POD (1:10,000 dilution; Jackson ImmunoResearch Laboratories), monoclonal mouse anti-α-tubulin (1:1000 dilution; clone B-5-1-2) and goat anti-mouse POD (1:10,000 dilution; Jackson ImmunoResearch Laboratories). All antibodies were diluted in 1% (w/v) BSA in PBS. The signal was captured with a blot scanner (C-DiGit) and quantified using Image Studio Digits version 4.0 (Licor).

### Immunofluorescence assay of oocytes

Three days after RNA injection, oocytes were fixed by incubation with 4% (v/v) paraformaldehyde in PBS for 4 h at room temperature. Fixed cells were washed three times with 3% (w/v) BSA in PBS, followed by permeabilizing with 0.05% (v/v) Triton X-100 in PBS for 60 min. After washing three times with 3% (w/v) BSA in PBS, the oocytes were incubated with rabbit anti-PfCRT antiserum (1:500 dilution) overnight at 4 °C. After washing again for three times, the secondary antibody, anti-rabbit Alexa Fluor 546 (dilution 1:1000), was added and allowed to incubate for 45 min. After 3 subsequent washing steps, the antibody serving as an internal control, wheat germ agglutinin Alexa Fluor™ 633 (5 µg/ml), was added and the oocytes were incubated for 10 min at 4 °C. After 3 more washing steps, the oocytes were analyzed by fluorescence microscopy. Images were taken with a Zeiss LSM 510 confocal microscope and processed with the Fiji program.

### Modeling of PfCRT^Dd2 in the open-to-vacuole conformation

The system was built, using the cryo-EM structure of the PfCRT isoform from the CQ-resistant *P. falciparum* strain 7G8, PDB ID 6UKJ[27], as a reference. The CharmmGUI[56] web server was used to setup the system with PfCRT embedded in a lipid bilayer by performing mutagenesis of residues to match the sequence of PfCRT^Dd2. Residues 114-122 were missing in the cryo-EM structure, and were modeled, using the GalaxyFill algorithm[57] implemented in CharmmGUI. A quality assessment of the final model is provided in Supplementary Table 3.

### Modeling of PfCRT^Dd2 in the occluded conformation

Modeling of the occluded conformation of PfCRT^Dd2 was performed following a procedure similar to the one described by Coppée et al. [38]. A model of the occluded conformation was built using homology modeling, the web server Phyre2[58] and the crystal structure from PDB ID 5Y79 (triose-phosphate/phosphate translocator)[37] as a template. This structure was chosen because it is the structure in the occluded state with highest sequence identity to PfCRT^Dd2. The sequence identity between PfCRT^Dd2 and the protein in PDB 5Y79 is 15%. The sequence alignment performed by Phyre2 prior modeling is shown in Supplementary Fig. 7. Despite a low sequence identity between PfCRT and the triose-phosphate/phosphate translocator, a high confidence score of 99.7% was obtained for the modeled PfCRT structure.

Confidence score larger than 90% indicates that the modeled protein adopts an overall folding similar to the template structure, and that the core of the protein is modeled at high accuracy (2-4 Å RMSD from the true structure)[58]. In the resulting model, residues 274-290 were missing. This region was included in the model using homology modeling, the program Modeller version 9.22[59,60] and the PDB ID 6UKJ as a template. A total of 50 models were generated and optimized, and the model with the lowest DOPE score was chosen. A quality assessment of the final model is provided in Supplementary Table 3.

### Modeling of PfCRT^Dd2 in the open-to-cytoplasm conformation

An initial model of the open-to-cytoplasm conformation of PfCRT^Dd2 was obtained using as basis the model of the open-to-cytoplasm conformation from PfCRT^7G8 published recently by Kim et al. [27] (model obtained from the authors). According to the original reference, in this model, the transmembrane helices were modeled using homology modeling, the structure from PDB 6UKJ as a reference and Maestro's Multiple Sequence Alignment tool, available in Schrödinger. This modeling strategy has been used previously for other membrane proteins[61,62], and takes advantage of the two-fold pseudo-symmetry of the transporter. This symmetry allows swapping the three-dimensional structure of the two repeated halves, enabling the use of the open-to-vacuole structure (PDB 6UKJ) to model the open-to-cytoplasm structure.

The loops and other missing segments connecting the transmembrane helices (residues 81-88, 110-125, 148-154, 169-180, 198-214, 235-251, 269-322, 336-344, 364-381) were missing. They were included in the model using the program Modeller version 9.22[59,60]. Including all the missing regions simultaneously in the model was challenging. Initial tests were done to include the missing regions without a template structure, but this led to the formation of knots in the model. Models without knots and with improved quality were achieved when the missing regions were modeled in two consecutive steps, using as template the model of the occluded conformation (described above), and with removal of the C-terminal (residues 362-402) from the modeling. The largest region missing, residues 269-322, was modeled in the first step, and then the remaining missing residues were modeled in the second step. A total of 200 models were generated and optimized, and the model with the lowest DOPE score was chosen. A quality assessment of the final model is provided in Supplementary Table 3.

### Simulation of PfCRT^Dd2 in different conformations

For each conformation (open-to-vacuole, occluded, open-to-cytoplasm) a total of 9 independent MD simulations of 200 ns were performed: 3 with residue E207 deprotonated, 3 with residue E207 protonated, and 3 with residue E207 mutated to His. The compositions of the different systems in terms of simulation box dimensions, total number of atoms, total number of water molecules, salt concentration, lipid composition is provided in Supplementary Table 7. Prior to membrane embedding of the protein, the protonation states of the residues at pH 6 were determined using Propka version 3.5[63–65], as implemented in the program pdb2pqr version 2.1.1[66,67]. The CharmmGUI[56] web server was used to setup the system with PfCRT embedded in a lipid bilayer. CharmmGUI generated topology files and coordinates for simulations with the Gromacs software package. A complex membrane consisting of cholesterol, 1-palmitoyl-2-oleoyl-sn-glycero-3-phosphocholine (POPC) and 1-stearoyl-2-oleoyl-phosphatidylethanolamine (SOPE) in the ratio 3:4:3 was built to represent the main components of *Xenopus laevis* oocyte membranes[68], employed in the experiments. The system was solvated with TIP3P[69] water molecules with a margin of at least 10 Å and Na+ and Cl− ions were added to ensure system neutrality at an ion concentration of 150 mM. Parameters to describe the protein and the lipids were obtained from the AMBER FF19SB[70] and the AMBER Lipid17[71] force fields, respectively.

MD simulations were carried out using GROMACS 2020[72]. The system obtained from CharmmGUI was energy minimized using positional restraints on the heavy atoms of the protein (4000 kJ/mol/nm² over backbone atoms, 2000 kJ/mol/nm² over side chain atoms) and membrane (1000 kJ/mol/nm²). The steepest descent method was used, and energy minimization was stopped after 5000 steps. The system was then equilibrated before the production run, using the standard six-step protocol suggested by CharmmGUI. The protocol includes two simulations with an NVT ensemble, followed by four simulations with an NPT ensemble. First, the system was heated to 300 K using the Berendsen thermostat[73]. Then, positional restraints over the protein (2000 kJ/mol/nm² over backbone atoms, 1000 kJ/mol/nm² over side chain atoms) and lipid heavy atoms (400 kJ/mol/nm²) were reduced. Next, the pressure was equilibrated to 1 bar using the Berendsen barostat[73] and positional restraints over the protein heavy atoms (1000 kJ/mol/nm² over backbone atoms, 500 kJ/mol/nm² over side chain atoms) were further reduced. After temperature and pressure equilibration, additional steps were performed to reduce the positional restraints over the protein (500, 200, 50 kJ/mol/nm2 over backbone atoms, 200, 50, 0 kJ/mol/nm² over side chain atoms) and lipid heavy atoms (200, 40, 0 kJ/mol/nm²). In the production runs of 200 ns, temperature coupling was achieved with the Nose-Hoover thermostat[74,75] and a time constant of 1 ps, and pressure coupling was achieved with the Parrinello-Rahman barostat[76,77] with a compressibility of $4.5\,10^{-5}\,bar^{-1}$ and a time constant of 5 ps. A time step of 2 fs was used. Covalent bonds to hydrogen in the solute were constraint using the LINCS algorithm[78]. Bond lengths for the solvent were constrained using the SETTLE algorithm[79]. Van-der-Waals forces were computed using a cutoff of 1.2 nm. Electrostatic forces were calculated using the particle mesh Ewald (PME) method[80,81] with a real-space cutoff of 1.2 nm, PME order of four, and a Fourier grid spacing of 1.2 Å.

### Docking of CQ to PfCRT$^{Dd2}$ in different conformations

Protein structures were collected with a frequency of 10 ns from the last 150 ns of each of the three replicas of each conformation (open-to-cytoplasm, occluded, open-to-cytoplasm) and each protonation state of E207, resulting in 45 protein structures for each conformation. Then, the protein structures were clustered using the method gromos[82], as implemented in Gromacs, the root mean squared deviation (RMSD) of residues 140-154 and 204-211, and a cutoff of 0.15 nm, 0.18 nm and 0.22 nm for the open-to-vacuole, occluded and open-to-cytoplasm conformations, respectively. The centers of the clusters (Supplementary Table 2) were used for docking. Docking of CQ to the cluster centers of all three conformations of PfCRT$^{Dd2}$ with different protonation states of E207 was performed with AutoDock Vina[83]. An exhaustiveness level of 8 and a cubic grid with a spacing of 0.375 Å, 80 points along each edge and center around the alpha carbon of E207 were used for docking. Bond rotations were allowed in the ligand, while the protein structures were kept rigid. For each cluster center, 20 ligand poses were generated by docking, and the top scoring pose (irrespective of the binding mode) was selected.

### Data analysis of MD simulations

The secondary structure of PfCRT throughout the MD simulations has been assigned with the DSSP analysis, performed using MDTraj[84]. The root mean squared deviation (RMSD) of the alpha carbons of all residues or of the alpha carbons of the transmembrane helices were calculated using GROMACS[72]. The residues that were part of one of the ten transmembrane helices of PfCRT at the beginning of the simulation (residues 49–57, 60–85, 90–113, 127–150, 153–173, 181–204, 211–236, 242–265, 279–290, 315–340, 342–361, 377–403 in the open-to-vacuole conformation, residues 54–86, 90–110, 129–150, 153–175, 182–201, 211–238, 246–264, 276–286, 317–339, 345–366, 375–397 in the occluded conformation, residues 61–85, 90–106, 130–134, 136–150, 156–168, 182–196, 212–230, 254–265, 276–288, 316–328, 346–352 in

the open-to-cytoplasm conformation) were listed as residues of the transmembrane helices and used for RMSD calculation. The root mean square fluctuation (RMSF) was calculated for the alpha carbons of all residues. For the calculation of distances involving hydrogen bond donors and acceptors, the oxygen atoms of the carboxylic group of E207, E372, D137, and D329, the NZ atom of K80, the ND atom of N84, the OG atom of S227, and the OG atom of T333 were considered. Distances were calculated using Gromacs[72].

### Statistical analysis of competition kinetics

Sixteen different kinetic models of drug competition were adopted from Segel[85]. For an in-depth derivation, readers are referred to this textbook. The Supplementary Information includes the kinetic equations describing each of the 16 competition models, along with the discriminatory statistical analysis based on the Akaike information criterion difference ($\Delta AIC_c$) and the Akaike weight[39,40].

### Statistics and reproducibility

Statistical analyses were performed using Sigma Plot (v.14.5, Systat) software. Statistical significance was determined using the Holm–Šidák one-way ANOVA test, the two-tailed Mann–Whitney test, the two-tailed Student's $t$ test, or the two-tailed $F$-test where appropriate. $p$ values < 0.05 were considered significant. The number of biologically independent samples is indicated in the main text and/or the figure legends. If data from biologically independent samples were averaged, then the mean ± SEM is shown. Biologically independent samples are defined as experiments using oocytes from different frog. Per condition and biologically independent sample, at least 10 oocytes were investigated.

### Reporting summary

Further information on research design is available in the Nature Portfolio Reporting Summary linked to this article.

## Data availability

All data supporting the findings of this study are available within the article and its supplementary information files, or available from the authors upon request. Source data and the structural models of PfCRT$^{Dd2}$ underlying this article are provided via the Dryad Digital Repository at https://doi.org/10.5061/dryad.2z34tmprm and the Zenodo Digital Repository at https://zenodo.org/record/8042801, respectively[86,87]. Experimental source data are also provided as a Source Data file. The reference protein structures used in this work are available from the protein data bank; 6UKJ [88], and 5Y79 [89]. Source data are provided with this paper.

## Code availability

The python code used for statistical analysis of the competition kinetics is provided via the Zenodo Digital Repository at https://doi.org/10.5281/zenodo.8037512 [90].

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

## Acknowledgements

The authors thank Marina Müller and Atdhe Kernaja for excellent technical assistance, and Augusto Masetti for his help with the Python code. We thank Rebecca Wade for advice, David Fidock for the rabbit anti-PfCRT antibody, and Jonathan Kim for the modeled open-to-cytoplasm conformation of PfCRT[7G8]. M.L. thanks the State of Baden-Württemberg for funding part of the work. A.N.-A. and F.S. acknowledge funding

from Deutsche Forschungsgemeinschaft under the Germany's Excellence Strategy—EXC 2008/1-390540038—UniSysCat. For the publication fee we acknowledge financial support by Deutsche Forschungsgemeinschaft within the funding programme "Open Access Publikationskosten" as well as by Heidelberg University.

## Author contributions

F.B., C.P.S., G.P., A.N.-A. and M.L. designed the study. F.B., G.M.G., B.P., F.S., A.N-A. performed experiments. F.B., G.M.G., C.P.S., B.P., F.S., A.N-A., G.P. and M.L. analyzed results. M.L. wrote the manuscript with the help of F.B. and A.N-A. All authors participated in discussion and manuscript editing.

## Funding

## Competing interests

The authors declare no competing interests.
