## [Peer Review File · Nature Communications]

pH-dependence of the Plasmodium falciparum chloroquine resistance transporter is linked to the transport cycleREVIEWER COMMENTS

Reviewer #1 (Remarks to the Author):

PfCRT, a transporter localized in the membrane of the digestive vacuole (DV), is a central mediator of resistance to antimalarials – chloroquine (CQ) and other chemically related 4-aminoquinolines. This manuscript by Berger et al. describes how the PfCRT CQ-resistant Dd2 isoform detects changes in pH during the efflux of CQ from the DV. Data were generated using alanine-scanning mutagenesis combined with drug uptake assays and molecular dynamics simulations. The study provides intriguing insights into the mechanistic basis of pH-dependent transport by the PfCRT Dd2 isoform. The authors have identified E207 as a potential pH-sensor and predict this residue to interact with neighboring ones in a pH-dependent manner facilitating the transition from the open-to-DV to the open-to-cytosol conformation. This work is certainly comprehensive and informative, and I think that it would be a great addition to Nature Communications.

I have a few specific suggestions/questions for the authors to consider:

- Is E207 conserved among other Plasmodia species? Is there any PfCRT drug resistant isoform that carries a mutation in residue 207?
- The authors implemented both alanine-scanning mutagenesis and functional assays to identify E207A as pH-independent. Can this approach be implemented in identifying a proton binding site in the TM region?
- In Supplementary Table 2, there seems to be a shift in pKa for D329 amongst distinct conformations. It would be interesting to see the functional outcome when D329 is mutated.
- There seems to be a difference between CQ and PPQ in R-value against transport activity at pH 6.0 for PfCRT Dd2. What would be an explanation for the difference in pH-dependence between CQ and PPQ?
- Based on the reported observations, what is the author's postulation on how PfCRT transitions back to the open-to-vacuole from the open-to-cytosol conformation?
- E207A is pH-independent while maintaining 44% of the transport activity at pH 6.0 compared to that of PfCRT Dd2. There seems to be an additional factor that drives drug transport. Can the authors elaborate more on why they think there is a residual transport activity?
- For Figure 3, to improve clarity, could the authors color/label TM helices?
- For Supplementary Fig. 4, it would be helpful if the TM domains were labeled for residues that are interacting with CQ.
- There might be a typo on page 15 line 352 where it should be R374 and not R373.

Reviewer #2 (Remarks to the Author):

The PfCRTDd2 transporter from the malaria parasite Plasmodium falciparum transports oligopeptides resulting from the digestion of haemoglobin into the digestive vacuole of the malaria parasite.

Chloroquine (CQ - malaria drug) resistant PfCRT mutants contain a number of mutations that enable PfCRT to expel CQ from the vacuole. i.e. PfCRT acts as in two directions. It transports oligopeptides into digestive vacuoles whereas it transports CQ out of the vacuole.

PfCRT operates at only 60% of its full velocity at the pH of the vacuole. This means that the selective pressure exerted by CQ was able to introduce mutations granting PfCRT the ability to

transport CQ out of vacuoles, but was unable to overcome the pH sensitivity of PfCRT.

In this study the author utilize an alanine scanning mutational approach to identify residues that regulate the pH sensitivity of PfCRT. Using an elegant screen wherein the relative transport rate at pH4.5 and pH6 was plotted against the rate at pH 6 they identify a number of residues that when mutated to alanine display altered pH sensitivity. The authors focus the remainder of the study on E207 which displays zero pH sensitivity when mutated to Alanine. Using extensive biophysical characterization coupled with molecular dynamics modeling and selected additional mutations, the authors present a model wherein E207 interacts with K80 to stabilize the conformation of the transporter in the "cytosol facing" state. When mutated to any other amino acid incapable of forming hydrogen bonds with K80, PfCRT becomes pH insensitive.

As such the study advances our knowledge on the relationship between structure and function in PfCRT, which plays a critical role in drug resistance in the malaria parasite. The manuscript is written in a clear language and does an excellent job at taking the reader through even difficult concepts and terms. The data quality is also high and I highly enjoyed the integration of wet lab data with structural and modeling data/approaches.

I have the following comments:

1) Of a technical nature, the extended kinetics study conducted in Figure 4 is claimed to measure initial transport rates. But in fact the assays are 1 hour long which is far from being initial transport rates. To measure initial transport rates one would have to use electrophysiology which measures transport rates (live). If the transporter is not electrogenic one would at the very least show a timecourse assay wherein one chooses an assay time wherein the increase in uptake is linear. In our experience at one hour the uptake into oocyte can have reached a stationary phase which can distort measurements of higher concentration uptake. I can see that the authors refer to four previous studies in the methods regarding how the oocyte experiments were done. But it is not clear whether time-course assays have been done there. So either the authors have to address this issue clearly by referring to previous studies or include a time course assay in the supplementary data wherein they show linear uptake at the one hour mark.

2) I am always curious when I read studies using *Xenopus* oocytes on transporters sitting in intracellular compartments (in their natural environments), whether the authors have any reflections on the direction of the transporter? It is clear that PfCRT is capable of importing CQ into the oocytes. i.e. it is clearly in the plasma membrane. However, considering that the authors are measuring the effect of pH on E207, then it is critical for the main conclusions that this residue is actually facing the extracellular side of the oocytes. Can the authors substantiate that this is indeed the case? I mean with no information on whether the transporter is coupled it could just be bidirectional and the E207 could be sitting inside the oocyte shielded from the pH changes outside the oocyte.

2) I am also curious whether PfCRT has ever been tested for electrogenicity? either when transporting oligopeptides or CQ? This could be addressed briefly in the results section to argue for the choice of the cumulative assay for investigating the kinetics of PfCRT.

3) I was disappointed that the authors did not test for oligopeptide transport. I almost cannot fathom that they didn't, but perhaps kept it out for good reasons? It seems to me that the question about why selective pressure did not overcome pH sensitivity could be due to opposing selective pressure exerted by reduced transport of oligopeptides (also see comment below).

4)

If I understand the authors correctly: at lower pH, E207 becomes protonated and is therefore not able to interact/form a hydrogen bond with K80. According to the model this means that the lower pH causes LACK of stabilization in the "facing cytosol state". This confuses me because you are testing transport of CQ. Shouldn't you then see more transport of CQ as the transporter might have an easier time/higher probability of facing the vacuole (extracellular oocyte) space from where it is binding CQ for translocation out of vacuole (into oocyte)? I would understand it, if you

were measuring oligopeptide transport where the stabilization in the "facing cytosol" state would be necessary for binding the oligopeptide. Or is it because you E207 's hydrogen binding to K80 is necessary for completion of (or progression through) the transport cycle? in such case the E207A in effect represents a pH insensitive but poorer version of PfCRT. Hence, the selective pressure of CQ would actually serve to preserve the pH sensitivity of PfCRT since mutating it would yield even less transport rates. Considering the interesting question posed in the intro of the article, I think for the general reader (if I am correct in my understanding) that this aspect could be addressed in a more clear way at the end of the discussion. The authors do an excellent job throughout the paper to explain difficult concepts but at the end I think the main message drowns a bit.

Reviewer #3 (Remarks to the Author):

Comments:

In this manuscript, Berger et al. report on structure-based study of pH-dependent conformational dynamics of chloroquine resistance transporter. Authors have performed mutagenesis analysis, transport and kinetic studies, and molecular dynamics simulations to investigate the molecular details of the role of titrable residues in pH sensing. Investigating the pH sensing in chloroquine binding is of high importance and results of this study will help us understand the subject better. However, the manuscript has several clear shortcomings which prevent me from recommending its publication in the current form. Here, my comments are mainly focused on the modeling and MD sections.

The primary concern is about the reliability of two models (cytoplasmic-facing and occluded models) used in this study. In this work, the authors used a cytoplasmic model that they abstained its initial construct from the Kim et al. authors. I cannot find neither here nor in Kim et al. paper how they built the model. From the very vague description in Kim et al., I guess the model was built based on the method developed by Lucy Forrest (Forrest et al., PNAS, 2007). If this is the case, the quality of a single model would not be in the level to use it for the docking or draw a conclusion based on side chain orientation (especially about the hydrogen network analysis in Supplementary Figure 5) and the pKa calculation. A better approach would have been to construct several models and relax the models using MD simulations and then use the models to run docking. For example, in modeling of both states, the authors re-built the loops near E207. How much the loop modeling affects the subsequent analyses.

Other points:

The authors select the snapshots from the MD simulation of the vacuole-facing model from every 10 ns of the last 50 ns. A better approach would have been to cluster the protein conformations and use their representative for docking.

In modeling the occluded state, it would be better to put the data about modeling in the paper rather than refereeing to another paper. For example, how much is the sequence similarity between two proteins and probably the sequence alignment in the supplementary section?

Figure 3b is hard to comprehend. It would be better to assign the main residues interacting with chloroquine. It is probably better to merge them with Supplementary Figure 4.

In Supplementary Table 2, there are residues with "not included in model". What does it mean? Are the models not complete?

Why are there two different pKa programs for different sections (pdb2pqr for assigning protonation states in the simulation and PropKa for the Supplementary Table 2)? Are there any differences between the results?

Reviewer #4 (Remarks to the Author):

The authors investigate the molecular basis for pH sensitivity of drug export by a mutant of the PfCRT transporter using functional measurements, mutagenesis as well as modeling (structures, docking, and MD). The primary result is that residue E207 is directly responsible for the pH-sensitivity of drug transport. The proposed mechanistic explanation suggests that E207 in its deprotonated (charged) form interacts with K80 and helps to stabilize the open-to-cytosol conformation and thus kinetically accelerates the part of the transport cycle from open-to-vacuole to open-to-cytosol. The work provides an answer to the open question if E207 is directly involved in proton (H⁺) transport or if its primary role is in conferring pH sensitivity. This paper posits that its primary role is in supporting the conformational transition through favorable interactions and that the pH sensitivity is an accidental side effect.

Overall, the work is fairly convincing with experiments and modeling/simulations supporting the conclusions, with the exception that the final hypothesis "We propose that [...] E207 acts [...] [for] accelerating the formation of the open-to-cytosol conformation." is mostly speculative.

I feel that the pKa predictions are used somewhat uncritically and I have some suggestions how the authors should address some of the known limitations.

Major comments

1) pKa predictions are really difficult and heuristic approaches such as PROPKA (even though useful to indicate tendencies and changes in the chemical environment) cannot be trusted at face value, especially when discussing functionally important residues such as E207. Probably the current gold standard for proteins are constant pH MD simulations but they are difficult. Instead, MD simulations should be performed for different protonation states of any relevant residue. In this case, MD simulations for (1) E207 deprotonated and (2) E207 protonated should be performed and compared.

2) Repeat the docking with different protonation states to check if this may have an influence (it probably will not but one cannot say with certain without doing the actual computation).

3) MD simulations of 100 ns are relatively short when it comes to assessing local conformational changes such as changes in hydrogen bonding pattern. At a minimum, perform 100-ns simulations in triplicate to assess robustness and variability.

Minor remarks

1) In the Introduction, make clearer what the thermodynamic driving forces for (1) the native oligopeptide transport and (2) the drug export are. It appears that at least drug export is H⁺ driven, but this is not clearly spelled out. It should also be made clear if the process happens (presumably?) as a symport process.

2) Make clear (Introduction and/or Discussion) if there is any physiological need for pH sensing.

3) What is the orientation of PfCRT in oocytes? Is it guaranteed to be in the "digestive vacuole" side (with E207) facing to the solution outside the oocyte, i.e. right-side out (cytosolic side facing cytosol) and how do the conclusions from the experiments depend on the orientation?

4) Given that kinetic measurements of E207A have been made, could the authors apply the same model selection process to those data with the same model selection criterion? This process should select a model without competition. This approach could be used to validate the general approach of model selection.

5) "E207 is not involved in proton transfer pathway" -- It's not clear to me that proton-driven transporters require proton-transfer chains. At least looking at secondary active transporters such as MFS transporters (eg LacY) and sodium/proton coupled transporters (eg TtNapA, PaNhaP), a single titratable residue that changes accessibility in the alternating access conformational transition may be sufficient; in other transporters (such as PiPT, MATE) two-step proton relay may be occurring. If the alternating access mechanism exposes the binding residue to solvent, transport chains for proton hops are not strictly necessary. Thus, although I agree with the authors' hypothesis that E207 is not directly involved in proton transfer, my reasoning would rather be that it is at the periphery and not at the center of the binding site around which the alternating access transition pivots.

6) In the discussion of the interaction network it was not entirely clear if the distances and networks were only computed from static models (from experiment or modeling). If only static models were used then they may contain spurious interactions. By basing such discussions on MD data one can appreciate the uncertainty better:

Show distributions of the discussed distances and for simulations with different protonation states explicitly modeled (in particular E207 (see above), potentially also for E372 and K80).

7) The authors present good evidence for the importance of the E207-K80 salt bridge. As an additional key validation I suggest the charge-swap double mutation E207K/K80E. If the salt-bridge is important, this may restore close to WT transport.

8) Line 287 "Having a pKa of 6.0, His can also be protonated".

This is fallacious reasoning because it invokes the solution pKa of His. The protein micro environment can modify the pKa. Without providing evidence that the actual pKa is ~6, this argument does not contribute anything to the discussion.

You could possibly create a mutant model and use PROPKA pKa predictions (similar to Suppl. Table 2) to get an idea of the pKa of E207H.

9) If only a hydrogen bond is important and not a salt-bridge then shouldn't E207Q should have same transport but no pH dependence? Was this mutant measured?

If E207Q had similar transport properties as WT but avoided the pH sensitivity, wouldn't this be a favorable mutant?

10) " $1 > \Delta AICc < 7$ " is very confusing

I suggest to express in words what is meant, e.g. "Models with $\Delta AICc$ between 1 and 7." or " $1 < \Delta AICc < 7$ ".

11) AIC was used to choose the best model but no model quality check of the best model itself was provided. E.g. are the residuals (RSS) Gaussian distributed, could cross-validation be performed (split data in training/test set)?

12) PROPKA is heuristic and indicative of the chemical environment of a residue but very dependent on detailed conformations. Running PROPKA on frames of MD trajectories can provide an indication of the variability and the distribution of pKa values.

13) Note that D329 also shows a substantial up-shift in pKa (5.2 to 9.0) according to PROPKA. If E207 is discussed based on the PROPKA results then the authors should also comment on D329 and discuss why (or why not) it may be relevant. Otherwise it appears as if results are being cherry-picked.

14) Provide some minimal quality assessment for structural models:
- CA RMSD (all residues and only transmembrane domains)
- secondary structure (e.g. DSSP), either per-residue time series or probability to maintain secondary structure

15) For reproducibility and transparency, make structural models of PfCRT publicly available, namely (1) open-to-cytoplasm and (2) occluded conformation. PDB files can be added to Supplementary Information or (better) in a public archive (with DOI) such as zenodo or figshare.

16) Add more details to modeling methods:

- For setting of protonation states at pH 6 PDB2PQR was used but state in main paper that really PROPKA was used (at the moment, this is only clear by reading Suppl Table 2) and cite PROPKA.

- add citations for thermostat and barostat algorithms, LINCS, SETTLE (for water)

- state treatment of non-bonded interactions (Coulomb and van der Waals/LJ) (with citation for eg Smooth PME)

17) Fig 1: sub-labels a-e are missing

Fig 4: sub-labels a-d are missing

Responses to reviewers' comments on manuscript NCOMMS-22-47067

We thank the reviewers for their encouraging comments and helpful suggestions.

Reviewer #1

PfCRT, a transporter localized in the membrane of the digestive vacuole (DV), is a central mediator of resistance to antimalarials – chloroquine (CQ) and other chemically related 4-aminoquinolines. This manuscript by Berger et al. describes how the PfCRT CQ-resistant Dd2 isoform detects changes in pH during the efflux of CQ from the DV. Data were generated using alanine-scanning mutagenesis combined with drug uptake assays and molecular dynamics simulations. The study provides intriguing insights into the mechanistic basis of pH-dependent transport by the PfCRT Dd2 isoform. The authors have identified E207 as a potential pH-sensor and predict this residue to interact with neighboring ones in a pH-dependent manner facilitating the transition from the open-to-DV to the open-to-cytosol conformation. This work is certainly comprehensive and informative, and I think that it would be a great addition to Nature Communications.

Thank you very much for your support!

I have a few specific suggestions/questions for the authors to consider:

- Is E207 conserved among other Plasmodia species? Is there any PfCRT drug resistant isoform that carries a mutation in residue 207?

We thank the reviewer for these questions. E207 is indeed conserved among most Plasmodium species, with the exception of the four rodent malaria parasites, *P. berghei*, *P. chabaudi*, *P. vinckei* and *P. yoelii* (according to www.plasmodb.org as of April 2023) (new supplementary Fig. 4a). Interestingly, the CRT orthologues of the rodent malaria parasites cluster far away from PfCRT in a bootstrap analysis (new supplementary Fig. 4b). It is tempting to speculate that the evolutionary distance results from adaptive processes that have led to a differentiation of the physiological functions exerted by CRT and the mechanisms regulating CRT activity. Unfortunately, very little is known about the natural substrate of CRT in other Plasmodium species. There is, however, evidence that the mode of action of chloroquine and the mechanism of resistance varies among Plasmodial species and may involve mutational changes in factors other than CRT, such as in the digestive vacuolar amino acid transporter AA1. We have included a short paragraph in the Results and Discussion section on the conservation of E207.

“E207 is present in all published PfCRT isoforms and it is also conserved among most Plasmodium species, with the exception of the four rodent malaria parasites, *P. berghei*, *P. chabaudi*, *P. vinckei* and *P. yoelii* (according to www.plasmodb.org as of April 2023) (supplementary Fig. 4a). Interestingly, the CRT orthologues of the rodent malaria parasites cluster far away from PfCRT in a bootstrap analysis (supplementary Fig. 4b), suggesting divergent physiological functions and, possibly, pH-regulation.” (Results section, end of page 7)

“E207 is conserved among most Plasmodium species, with the exception of the four rodent malaria parasites, *P. berghei*, *P. chabaudi*, *P. vinckei* and *P. yoelii* (supplementary Fig. 4). The evolutionary distance between PfCRT and its orthologues in rodent malaria parasites might result from adaptive processes that have led to a differentiation of the physiological functions exerted by CRT and the mechanisms regulating CRT activity. Unfortunately, very little is known about the natural substrate of CRT in other Plasmodium species. There is, however, evidence that the mode

of action of chloroquine and the mechanism of resistance varies among Plasmodial species and may involve mutational changes in factors other than CRT, such as in the digestive vacuolar amino acid transporter AA1^{45,46}.” (Discussion, end of page 15)

• The authors implemented both alanine-scanning mutagenesis and functional assays to identify E207A as pH-independent. Can this approach be implemented in identifying a proton binding site in the TM region?

We thank the reviewer for this question. The screen conducted by us and the data presented in the manuscript can also inform on possible candidates for proton transfer. Criteria for candidates include, as deduced from studies conducted in other systems: i) a titratable residue, frequently Glu or Asp; ii) presence in a transmembrane domain; iii) location in the center of the channel; iv) ability to engage in inter-helical H-bonding, frequently with Ser or Thr for coupling protonation dynamics with conformational changes; and v) a strong effect on transport activity. Applying these criteria to PfCRT^{Dd2} and CQ transport identified D137 as a promising candidate. We have highlighted both residues in the revised Fig. 1d. However, further extensive experiments would be needed to test the role of these two residues in the transport function of PfCRT. Regardless, given the query by this reviewer and reviewer 4, we have included a paragraph in the results and the discussion, introducing both residues as plausible candidates for proton-shuttling. However, we do not wish to overemphasize these results given their preliminary nature.

“To identify more plausible candidates for proton-transfer, we re-examined the alanine mutants, but this time with a different set of criteria in mind. As shown in other systems, residues involved in proton-binding and proton-transfer are titratable and frequently consist of Glu and Asp; they are present in a transmembrane domain and are located in the center of the channel; they are able to engage in inter-helical H-bonding, frequently with Ser or Thr, for coupling protonation dynamics with conformational changes; and replacing them strongly affects the transport activity⁴⁴⁻⁴⁶. Applying these criteria identified D137 and D329 as initial candidates. Both residues are located in transmembrane domains (TM3 and TM8, respectively; Supplementary Fig. 2) and replacing them by the isosteric amino acid Asn drastically reduced the transport activity for CQ at pH 6.0 (Fig. 1d). Both residues are conserved among *Plasmodium* spp (Supplementary Fig. 4a) and structural models placed both residues in the center of the channel (Supplementary Fig. 12). MD simulations revealed that D137 forms an inter-helical hydrogen bond with S227 (residing on TM 6; distance of 1.85 Å) in the open-to-vacuole and the occluded conformation (Supplementary Figs. 12a and b, and 13), consistent with a previous report⁴⁷. In comparison, D329 engages in intra-helical hydrogen bonding - with T333 (like D329 also residing on TM 8; distance of 2.26 Å) (Supplementary Figs. 12c and d, and 13). These initial findings would favor D137 as a plausible candidate for proton shuttling, although further experiments are needed to clarify the role of D137 and also that of D329 in the transport cycle.” (results, page 11 and 12).

“A more promising candidate for proton binding and transfer is residue D137 on TM 3. Inserting a negatively charged residue in a transmembrane domain is energetically unfavorable ($\sim 4-5$ kcal mol⁻¹) and is, therefore, avoided unless the residue plays a functional role, e.g., in proton-transfer⁴⁵. To couple protonation dynamics with conformational changes, negatively charged residues frequently engage in inter-helical hydrogen bond formation via their carboxylate group with the hydroxyl group of Ser or Thr⁴⁵. This particular inter-helical carboxyl-hydroxyl interaction appears to be a hallmark of a proton-transfer mechanism in a membrane transporter⁴⁵. MD simulations indicated that D137 forms inter-helical hydrogen bonds with S227 on TM 6 in both the open-to-vacuole and occluded conformations, supporting its involvement in proton transfer (Supplementary Fig. 12). Additionally, substitution of D137 with the isosteric amino acid Asn greatly reduced CQ transport activity, consistent with the role of D137 in proton shuttling (Fig. 1d). While initially considering D329 as a potential candidate for proton transfer, we favored an

alternative function due to its absence of inter-helical hydrogen bonding with Ser or Thr, such as an involvement in substrate binding ³⁶. Further experiments, including constant pH MD simulations, are necessary to fully elucidate the functions of both residues in the transport cycle of PfCRT^{Dd2}.” (discussion, page 19).

- In Supplementary Table 2, there seems to be a shift in pKa for D329 amongst distinct conformations. It would be interesting to see the functional outcome when D329 is mutated.

D329 is indeed a very special residue (see also above). D329 was mutagenized in our initial scan to the isosteric amino acid Asn, resulting in a carrier with drastically reduced CQ transport activity (see Fig. 1d). However, the mutant was not highlighted in the original Fig. 1d since it maintained some degree of pH-sensitivity. Inspired by your comment above, we re-analyzed the alanine scanning mutants for residues that meet the criteria of a pH-shuttle. D329 was one of the two residues initially considered, but was later dismissed due to a lack of detectable inter-helical bond formation with Ser or Thr.

Willems et al. (2023) have recently proposed, based on MD simulations, that D329 plays a role in CQ binding. We mention D329 in the revised manuscript, but prefer to interpret the data cautiously. See also comment above.

- There seems to be a difference between CQ and PPQ in R-value against transport activity at pH 6.0 for PfCRT Dd2. What would be an explanation for the difference in pH-dependence between CQ and PPQ?

We also noticed this. While we can confirm that the differences between CQ and PPQ in R-value are reproducible, we have yet to find a plausible explanation other than blaming it on the pKa values (5.39, 5.72, 6.24, 6.88), which are in the operative range of PfCRT, unlike those of CQ, QN and QD. As a consequence, the abundance of the various species of PPQ differ between pH 4.5 and pH 6.0, which, in turn, may affect the overall transport activity. To clarify this issue, we have added the following paragraph:

“In the case of PPQ, the R-value could only be approximated since the pKa values of PPQ are in the operative range of PfCRT (5.39, 5.72, 6.24, 6.88) ³³. As a consequence, the distribution of the various species likely differs between pH 4.5 and pH 6.0, which, in turn, may affect the overall transport activity, assuming distinct kinetic properties for each species.” (Results, page 7).

- Based on the reported observations, what is the author’s postulation on how PfCRT transitions back to the open-to-vacuole from the open-to-cytosol conformation?

This is a good question to which we have no informed answer. We can only speculate that recycling of the transporter is induced by the release of the substrate(s). Alternatively, proton binding, possibly at the digestive vacuolar site, might reorient PfCRT from the open-to-cytoplasm to the open-to-vacuolar conformation. Examples of both mechanisms can be found in the literature (Parker et al., 2017; Bozzi et al., 2020). But again, this is all speculative and further experiments, in particular high-resolution structural information combined with mutational analyses, would be required to resolve this issue.

- E207A is pH-independent while maintaining 44% of the transport activity at pH 6.0 compared to that of PfCRT Dd2. There seems to be an additional factor that drives drug transport. Can the authors elaborate more on why they think there is a residual transport activity?

In our study, we show that the pH sensing residue E207 plays a critical role in the transport cycle by interacting with specific residues in the occluded and open-to-cytoplasm state. We propose that these interactions accelerate the transport cycle and, hence, the transport activity. According to this model, the residual activity seen in the E207A mutant represents the unaccelerated transport cycle that may be driven by proton-cotransport or be induced by CQ binding. We have amended the test as follows:

“On the basis of our findings, we postulate that E207 plays a critical role in the drug-transporting function of PfCRT^{Dd2} by accelerating progression through the transport cycle via salt bridge formation with K80 as the carrier transits from the open-to-vacuole conformation to the occluded state. In comparison, the E207A variant is unable to interact with K80 and, thus, represents a pH-insensitive but poorer version of PfCRT^{Dd2} (Fig. 7). Its residual activity may be driven by proton-cotransport^{25,26} or be induced by CQ binding.” (Discussion, page 19)

- For Figure 3, to improve clarity, could the authors color/label TM helices?

We have improved the quality of all figure related to MD simulations and docking. As suggested by the reviewer, TM helices were colored and labeled in Fig. 3a. TM helices were also labeled in Fig. 3b, but not colored, to avoid a decrease in clarity. In general, we have improved the quality of all figures related to docking and molecular dynamics simulations.

- For Supplementary Fig. 4, it would be helpful if the TM domains were labeled for residues that are interacting with CQ.

Supplementary Fig. 4 has been removed and replaced by Supplementary Fig. 6, which shows the predicted binding modes of CQ for all clusters or groups of protein configurations obtained from molecular dynamics simulations. We followed the suggestion of the reviewer, and labeled the TM helices in the figures to improve clarity.

- There might be a typo on page 15 line 352 where it should be R374 and not R373. We apologize for the spelling mistake.

Reviewer #2

The PfCRT^{Dd2} transporter from the malaria parasite *Plasmodium falciparum* transports oligopeptides resulting from the digestion of haemoglobin into the digestive vacuole of the malaria parasite. Chloroquine (CQ - malaria drug) resistant PfCRT mutants contain a number of mutations that enable PfCRT to expel CQ from the vacuole. i.e. PfCRT acts as in two directions. It transports oligopeptides into digestive vacuoles whereas it transports CQ out of the vacuole. PfCRT operates at only 60% of its full velocity at the pH of the vacuole. This means that the selective pressure exerted by CQ was able to introduce mutations granting PfCRT the ability to transport CQ out of vacuoles, but was unable to overcome the pH sensitivity of PfCRT. In this study the author utilize an alanine scanning mutational approach to identify residues that regulate the pH sensitivity of PfCRT. Using an elegant screen wherein the relative transport rate at pH4.5 and pH6 was plotted against the rate at pH 6 they identify a number of residues that when mutated to alanine display altered pH sensitivity. The authors focus the remainder of the study on E207 which displays zero

pH sensitivity when mutated to Alanine. Using extensive biophysical characterization coupled with molecular dynamics modeling and selected additional mutations, the authors present a model wherein E207 interacts with K80 to stabilize the conformation of the transporter in the "cytosol facing" state. When mutated to any other amino acid incapable of forming hydrogen bonds with K80, PfCRT becomes pH insensitive.

As such the study advances our knowledge on the relationship between structure and function in PfCRT, which plays a critical role in drug resistance in the malaria parasite. The manuscript is written in a clear language and does an excellent job at taking the reader through even difficult concepts and terms. The data quality is also high and I highly enjoyed the integration of wet lab data with structural and modeling data/approaches.

We thank the reviewer for his/her encouragement and supportive comments.

I have the following comments:

1) Of a technical nature, the extended kinetics study conducted in Figure 4 is claimed to measure initial transport rates. But in fact the assays are 1 hour long which is far from being initial transport rates. To measure initial transport rates one would have to use electrophysiology which measures transport rates (live). If the transporter is not electrogenic one would at the very least show a timecourse assay wherein one chooses an assay time wherein the increase in uptake is linear. In our experience at one hour the uptake into oocyte can have reached a stationary phase which can distort measurements of higher concentration uptake. I can see that the authors refer to four previous studies in the methods regarding how the oocyte experiments were done. But it is not clear whether time-course assays have been done there. So either the authors have to address this issue clearly by referring to previous studies or include a time course assay in the supplementary data wherein they show linear uptake at the one hour mark.

The reviewer is absolutely right. It is a mistake on our part to refer to an initial transport activity when in fact we meant to say that our one-hour determinations are approximately in the linear uptake range. That this is the case has been independently confirmed by studies conducted in and Rowena Martin's and our lab (Bellanca et al., 2014; Martin et al., 2009; Summers et al., 2014). Irrespectively, we redid the time course experiments and present the additional data as a new Supplementary Fig. 17.

2) I am always curious when I read studies using *Xenopus* oocytes on transporters sitting in intracellular compartments (in their natural environments), whether the authors have any reflections on the direction of the transporter? it is clear that PfCRT is capable of importing CQ into the oocytes. i.e. it is clearly in the plasma membrane. However, considering that the authors are measuring the effect of pH on E207, then it is critical for the main conclusions that this residue is actually facing the extracellular side of the oocytes. Can the authors substantiate that this is indeed the case? I mean with no information on whether the transporter is coupled it could just be bidirectional and the E207 could be sitting inside the oocyte shielded from the pH changes outside the oocyte.

The reviewer raised an important concern. In a seminal study, Rowena Martin and her team established the oocyte system for functional studies on PfCRT (Martin et al., 2009). They showed that the topology of PfCRT is such that both the N- and C-terminus face the extracellular milieu (Martin et al., 2009; Shafik et al., 2020). They further demonstrated that PfCRT transports CQ in a manner dependent on a trans-membrane pH gradient – from the extracellular medium with a pH between 4.5 to 5.5 to the oocyte cytoplasm with a pH of ~ 7.2. Since then, the oocyte system has been widely used by Rowena Martin and us to study the function of PfCRT (Martin et al., 2009;

Shafik et al., 2020; Bakouh et al., 2017; Bellanca et al., 2014; Summers et al., 2014). Please note that, in the parasite, PfCRT-mediated CQ transport is from the lumen of the digestive vacuole with a pH of 5.2 to the cytoplasm with a pH of ~ 7.2. Both the N- and C-terminus are oriented towards the digestive vacuolar lumen (Kuhn et al., 2007). The orientation of E207 in the open-to vacuole conformation was taken from published structural data based on single particle imaging (Kim et al., 2019).

We have included the following statement:

“The direction of radio-labeled drug transport is here from the mostly acidic extracellular solution (pH 4.5 or 6.0, as indicated) into the oocyte cytosol (pH ~ 7.2), which corresponds to the efflux of drug from the acidic digestive vacuole (pH 5.2) into the parasite cytoplasm (pH 7.2). The topology of PfCRT in the oolemma is such that both the N- and C-terminus face the extracellular milieu^{3,22}. In the parasite, the N- and C-terminus face the lumen of the digestive vacuole²³. (Methods, page 21, second paragraph).

2) I am also curious whether PfCRT has ever been tested for electrogenicity? either when transporting oligopeptides or CQ? This could be addressed briefly in the results section to argue for the choice of the cumulative assay for investigating the kinetics of PfCRT.

We have previously shown that the addition of CQ elicits an inward rectifying current in voltage-clamped PfCRT^{Dd2}-expressing oocytes under conditions comparable to those described in the presented manuscript (Bakouh et al., 2017). The chloroquine-induced currents were sensitive to verapamil, a full mixed type inhibitor of PfCRT^{Dd2}. We explained the negative CQ-induced currents by the influx of net positive charges into the oocyte, which is consistent with PfCRT^{Dd2}-mediated CQ transport in association with protons. Thus, there is indeed initial evidence of PfCRT^{Dd2} being an electrogenic carrier of CQ. To clarify this issue, we have added the following statement:

“Furthermore, previous studies have shown that PfCRT-mediated CQ transport is electrogenic²⁴ and associated with a proton leak from the digestive vacuole^{25,26}. These findings led to the hypothesis of PfCRT functioning as a proton-coupled symporter.” (introduction, page 3).

3) I was dissatisfied that the authors did not test for oligopeptide transport. I almost cannot fathom that they didn't, but perhaps kept it out for good reasons? it seems to me that the question about why selective pressure did not overcome pH sensitivity could be due to opposing selective pressure exerted by reduced transport of oligopeptides (also see comment below).

Rowena Martin and her team recently published a study demonstrating pH-sensitivity of oligopeptide transport via the wild type and drug-resistance associated PfCRT isoforms, including the PfCRT^{Dd2} variant used in our study, in the heterologous oocyte system (Shafik et al., 2020). We now mention this finding in the introduction. We have not investigated oligopeptide transport via PfCRT and are currently unable to do so because of long delivery times for the custom-made radio-labeled oligopeptides. We have included the following statement in the manuscript:

“For example, the transport of oligopeptides and CQ is pH sensitive^{3,22}.” (Introduction, page 3)

4) If I understand the authors correctly: at lower pH, E207 becomes protonated and is therefore not able to interact/form a hydrogen bond with K80. According to the model this means that the lower pH causes LACK of stabilization in the "facing cytosol state". This confuses me because you are testing transport of CQ. Shouldn't you then see more transport of CQ as the transporter might have an easier time/higher probability of facing the vacuole (extracellular oocyte) space from where it is binding CQ for translocation out of vacuole (into oocyte)? I would understand it,

if you were measuring oligopeptide transport where the stabilization in the "facing cytosol" state would be necessary for binding the oligopeptide. Or is it because you E207 's hydrogen binding to K80 is necessary for completion of (or progression through) the transport cycle? in such case the E207A in effect represents a pH insensitive but poorer version of PfCRT. Hence, the selective pressure of CQ would actually serve to preserve the pH sensitivity of PfCRT since mutating it would yield even less transport rates. Considering the interesting question posed in the intro of the article, I think for the general reader (if I am correct in my understanding) that this aspect could be addressed in a more clear way at the end of the discussion. The authors do an excellent job throughout the paper to explain difficult concepts but at the end I think the main message drowns a bit.

Your remarks have been well received. The second model is the one we favor. In response to the reviewer's suggestion, we have re-written the last paragraph of the discussion:

"On the basis of our findings, we postulate that E207 plays a critical role in the drug-transporting function of PfCRT^{Dd2} by accelerating progression through the transport cycle via salt bridge formation with K80 as the carrier transits from the open-to-vacuole conformation the occluded state. In comparison, the E207A variant is unable to interact with K80 and, thus, represents a pH-insensitive but poorer version of PfCRT^{Dd2} (Fig. 7). Its residual activity may be driven by proton-cotransport^{25,26} or be induced by CQ binding. Similarly, if the carboxyl side chain of E207 is protonated, E207 would no longer be able to serve as a proton acceptor, which, in turn, would prevent it from interacting with K80, resulting in a reduced transport activity (Fig. 7). The additional proton may also cause spatial problems during the transport process. It is, therefore, advantageous for the parasite to preserve the pH sensitivity of PfCRT^{Dd2} and operate below maximal turn-over rates at the pH of ~ 5.2 of the digestive vacuole since mutating E207 would yield even less transport activity." (Discussion, page 19)

Reviewer #3

Comments:

In this manuscript, Berger et al. report on structure-based study of pH-dependent conformational dynamics of chloroquine resistance transporter. Authors have performed mutagenesis analysis, transport and kinetic studies, and molecular dynamics simulations to investigate the molecular details of the role of titrable residues in pH sensing. Investigating the pH sensing in chloroquine binding is of high importance and results of this study will help us understand the subject better. However, the manuscript has several clear shortcomings which prevent me from recommending its publication in the current form. Here, my comments are mainly focused on the modeling and MD sections.

We thank the reviewer for his/her insightful comments and suggestions and greatly appreciate the effort to help us improve your study.

The primary concern is about the reliability of two models (cytoplasmic-facing and occluded models) used in this study. In this work, the authors used a cytoplasmic model that they abstained its initial construct from the Kim et al. authors. I cannot find neither here nor in Kim et al. paper how they built the model. From the very vague description in Kim et al., I guess the model was

built based on the method developed by Lucy Forrest (Forrest et al., PNAS, 2007). If this is the case, the quality of a single model would not be in the level to use it for the docking or draw a conclusion based on side chain orientation (especially about the hydrogen network analysis in Supplementary Figure 5) and the pKa calculation. A better approach would have been to construct several models and relax the models using MD simulations and then use the models to run docking. For example, in modeling of both states, the authors re-built the loops near E207. How much the loop modeling affects the subsequent analyses.

We thank the reviewer for these insightful suggestions and have performed the docking experiments as recommended. We further provide now more detail on the experimental strategy in the methods section.

In particular, we described in more detail the modeling of the open-to-cytoplasm conformation. In summary, the initial model was obtained from Kim et al. According to the original reference, the transmembrane helices were modeled using homology modeling, the structure from PDB 6UKJ as a reference and Maestro's Multiple Sequence Alignment tool, available in Schrödinger. This modeling strategy has been used previously for other membrane proteins (Liao et al, 2012; Forrest, 2013), and takes advantage of the two-fold pseudo-symmetry of the transporter. This symmetry allows swapping the three-dimensional structure of the two repeated halves, enabling the use of the open-to-vacuole structure (PDB 6UKJ) to model the open-to-cytoplasm structure. The loops and other missing segments connecting the transmembrane helices were missing. They were included in the model using the program Modeller version 9.22. Including all the missing regions simultaneously in the model was challenging. Initial tests were done to include the missing regions without a template structure, but this led to the formation of knots in the model. Models without knots and with reasonable quality were achieved when the missing regions were modeled in two consecutive steps, using as template the model of the occluded conformation, and with removal of the C-terminal from the modeling. A quality assessment of the final model is provided in new Supplementary Table 3 and new Supplementary Figures 8 and 9.

In the revised version of the manuscript, we performed three independent 200-ns MD simulations of the three different conformations of PfCRT^{Dd2} (open-to-vacuole, occluded, open-to-cytoplasm) and two different protonation states for E207. We used the structures (or clusters of structures) from simulations for docking and pKa calculations. CQ did not show consistent interactions with E207 in the protonated or deprotonated forms (Figure 3, Supplementary Figure 6). The pKa values estimated for E207 tend to be lower for the occluded conformation, with E207 deprotonated (pKa range 2.34-5.27, Supplementary Table 4), in comparison with the occluded conformation, with E207 protonated (pKa range 4.19-6.78, Supplementary Table 4), the open-to-vacuole (pKa ranges of 3.75-5.18 and 3.21-5.60 for E207 protonated and deprotonated, respectively, Supplementary Table 4) and open-to-cytoplasm conformations (pKa ranges of 3.52-5.33 and 2.81-5.53 for E207 protonated and deprotonated, respectively, Supplementary Table 4).

Annex 1 (Table 1) shows how the pKa values change in the top 10 loop models (lowest DOPE scores) provided by Modeller for the occluded and open-to-cytoplasm conformations. For E207, the range of pKa values is 6.01-9.51 and 5.45-6.08 for the occluded and open-to-cytoplasm conformations, respectively.

Other points:

The authors select the snapshots from the MD simulation of the vacuole-facing model from every 10 ns of the last 50 ns. A better approach would have been to cluster the protein conformations and use their representative for docking.

We followed the suggestion made by the reviewer. We clustered the protein configurations obtained from MD simulations and docked CQ to the centers of each cluster. Results are shown in Figure 3, Supplementary Figure 6 and Supplementary Table 2. Clustering and docking was performed using the last 150 ns of three independent 200-ns MD simulations for six different systems: open-to-vacuole conformation (E207 protonated and deprotonated), occluded conformation (E207 protonated and deprotonated) and open-to-cytoplasm conformation (E207 protonated and deprotonated).

“Docking of CQ to PfCRT^{Dd2} in different conformations. Protein structures were collected with a frequency of 10 ns from the last 150 ns of each of the three replicas of each conformation (open-to-cytoplasm, occluded, open-to-cytoplasm) and each protonation state of E207, resulting in 45 protein structures for each conformation. Then, the protein structures were clustered using the method gromos⁸², as implemented in Gromacs, the root mean squared deviation (RMSD) of residues 140-154 and 204-211, and a cutoff of 0.15 nm, 0.18 nm and 0.22 nm for the open-to-vacuole, occluded and open-to-cytoplasm conformations, respectively. The centers of the clusters (Supplementary Table 2) were used for docking. Docking of CQ to the cluster centers of all three conformations of PfCRT^{Dd2} with different protonation states of E207 was performed with AutoDock Vina⁸³. An exhaustiveness level of 8 and a cubic grid with a spacing of 0.375 Å, 80 points along each edge and center around the alpha carbon of E207 were used for docking. Bond rotations were allowed in the ligand, while the protein structures were kept rigid. For each cluster center, 20 ligand poses were generated by docking, and the top scoring pose (irrespective of the binding mode) was selected.” (Methods, page 26)

In modeling the occluded state, it would be better to put the data about modeling in the paper rather than refereeing to another paper. For example, how much is the sequence similarity between two proteins and probably the sequence alignment in the supplementary section?

In the revised manuscript, we improved the description of the modeling of the occluded state. In summary, modeling was performed following a procedure similar to the one in Coppée et al. (2020) The model was built using homology modeling, the web server Phyre2 and the crystal structure from PDB ID 5Y79 (triose-phosphate /phosphate translocator) as a template. This structure was chosen because it is the structure in the occluded state with highest sequence identity to PfCRT. In the resulting model, residues 274-290 were missing. This region was included in the model using homology modeling, the program Modeller version 9.22 and the PDB ID 6UKJ as a template. A total of 50 models were generated and optimized, and the model with the lowest DOPE score was chosen. A quality assessment of the final model is provided in supplementary Table 3.

The sequence similarity between the template protein (PDB 5Y79) and PfCRT is 15%. The sequence alignment performed by Phyre2 prior modeling is shown in Supplementary Figure 7.

Figure 3b is hard to comprehend. It would be better to assign the main residues interacting with chloroquine. It is probably better to merge them with Supplementary Figure 4.

The main point of Figure 3b is to show that CQ does not have frequent interactions with E207 across the different binding modes obtained from docking. To keep clarity and enhance visualization, we opted for not showing other residues, apart from E207. For more information regarding CQ binding sites, we would kindly refer the reviewer to a recent publication by Willems et al. (2023). Their work is now cited the manuscript.

In Supplementary Table 2, there are residues with “not included in model”. What does it mean? Are the models not complete?

In the previous version of the manuscript, the models for the occluded and open-to-cytoplasm conformation were not complete. In the revised manuscript, the models for both conformations are complete, with the exception of a C-terminal region that was not modeled in the open-to-cytoplasm conformation, as detailed in Supplementary Table 3.

Why are there two different pKa programs for different sections (pdb2pqr for assigning protonation states in the simulation and PropKa for the Supplementary Table 2)? Are there any differences between the results?

We apologize for the confusion in the description of the program used to predict pKa values and protonation states. We used the program Propka version 3.5, as implemented in the program pdb2pqr version 2.1.1, to predict pKa values and assign protonation states. The description of the programs used is modified in the Methods section in the revised version of the manuscript (page 25).

Reviewer #4

The authors investigate the molecular basis for pH sensitivity of drug export by a mutant of the PfCRT transporter using functional measurements, mutagenesis as well as modeling (structures, docking, and MD). The primary result is that residue E207 is directly responsible for the pH-sensitivity of drug transport. The proposed mechanistic explanation suggests that E207 in its deprotonated (charged) form interacts with K80 and helps to stabilize the open-to-cytosol conformation and thus kinetically accelerates the part of the transport cycle from open-to-vacuole to open-to-cytosol. The work provides an answer to the open question if E207 is directly involved in proton (H⁺) transport or if its primary role is in conferring pH sensitivity. This paper posits that its primary role is in supporting the conformational transition through favorable interactions and that the pH sensitivity is an accidental side effect.

Overall, the work is fairly convincing with experiments and modeling/simulations supporting the conclusions, with the exception that the final hypothesis "We propose that [...] E207 acts [...] [for] accelerating the formation of the open-to-cytosol conformation." is mostly speculative.

I feel that the pKa predictions are used somewhat uncritically and I have some suggestions how the authors should address some of the known limitations.

We thank the reviewer for the offered help and have tried our very best to follow the reviewer's suggestions as close as possible within the limitations of our system.

Major comments

1) pKa predictions are really difficult and heuristic approaches such as PROPKA (even though

useful to indicate tendencies and changes in the chemical environment) cannot be trusted at face value, especially when discussing functionally important residues such as E207. Probably the current gold standard for proteins are constant pH MD simulations but they are difficult. Instead, MD simulations should be performed for different protonation states of any relevant residue. In this case, MD simulations for (1) E207 deprotonated and (2) E207 protonated should be performed and compared.

We thank the reviewer for the suggestion. Following the suggestion, we performed three 200-ns MD simulations for six different systems in the revised manuscript: open-to-vacuole conformation (E207 protonated, E207 deprotonated), occluded conformation (E207 protonated, E207 deprotonated) and open-to-cytoplasm conformation (E207 protonated, E207 deprotonated). In total 18 200-ns MD simulations were performed, resulting in an aggregate time of 3.6 μ s. The average, minimum and maximum pKa values for each system were calculated using multiple snapshots collected from the simulations, and are presented in the new Supplementary Tables S4. Interestingly, in the simulations we could observe the formation of hydrogen bonds between E207 and K80 in the occluded and open-to-cytoplasm conformations only when E207 is in the deprotonated state (new Figure 5, new Supplementary Figures 14).

2) Repeat the docking with different protonation states to check if this may have an influence (it probably will not but one cannot say with certain without doing the actual computation).

We followed the suggestion of the reviewer. The results for docking to conformations with different protonation states are shown in the new Figure 3 and new Supplementary Figure 6. We clustered the protein configurations obtained from MD simulations and performed docking of CQ using the center of each cluster. The binding modes of CQ are not the same for the same conformation and different protonation states of E207 due to the different configurations obtained from simulations. Most binding modes of CQ were far from E207, or established non-specific hydrophobic interactions with E207. No clear preferences of CQ for E207 in the protonated or deprotonated states could be detected.

3) MD simulations of 100 ns are relatively short when it comes to assessing local conformational changes such as changes in hydrogen bonding pattern. At a minimum, perform 100-ns simulations in triplicate to assess robustness and variability.

Following the suggestion of the reviewer, in the revised manuscript, we performed 3 replicas with a duration time of 200 ns each for the three different conformations of PfCRT, and two different protonation states of E207 for each conformation (new Supplementary Figure 14).

Minor remarks

1) In the Introduction, make clearer what the thermodynamic driving forces for (1) the native oligopeptide transport and (2) the drug export are. It appears that at least drug export is H⁺ driven, but this is not clearly spelled out. It should also be made clear if the process happens (presumably?) as a symport process.

We have amended the text to clarify this issue:

“Furthermore, previous studies have shown that PfCRT-mediated CQ transport is electrogenic²⁴ and associated with a proton leak from the digestive vacuole^{25,26}. These findings led to the hypothesis of PfCRT functioning as a proton-coupled symporter.” (Introduction, page 3)

2) Make clear (Introduction and/or Discussion) if there is any physiological need for pH sensing.

We thank the reviewer for this comment. A physiological function of the pH-sensitivity has not yet been established. In analogy with other transport systems (Gunshin et al., 1997), one may speculate that pH-sensitivity prevents an uncoupled proton leakage via PfCRT in case of an acidic overload in the digestive vacuole. To clarify this issue, we have added the following statement:

“If the pH-sensing ability of E207 does serve a physiological purpose, then it might prevent detrimental uncoupled proton leakage via PfCRT in case of an acid overload in the parasite’s digestive vacuole, as shown in other systems⁴⁴.” (Discussion, page 15)

3) What is the orientation of PfCRT in oocytes? Is it guaranteed to be in the "digestive vacuole" side (with E207) facing to the solution outside the oocyte, i.e. right-side out (cytosolic side facing cytosol) and how do the conclusions from the experiments depend on the orientation?

In a seminal study, Rowena Martin and her team established the oocyte system for functional studies on PfCRT (Martin et al., 2009). They showed that the topology of PfCRT is such that both the N- and C-terminus face the extracellular milieu (Martin et al., 2009; Shafik et al., 2020). They further demonstrated that PfCRT transports CQ in a manner dependent on a trans-membrane pH gradient – from the extracellular medium with a pH between 4.5 to 5.5 to the oocyte cytoplasm with a pH of ~ 7.2. Since then, the oocyte system has been widely used by Rowena Martin and us to study the function of PfCRT (Martin et al., 2009; Shafik et al., 2020; Bakouh et al., 2017; Bellanca et al., 2014; Summers et al., 2014). Please note that, in the parasite, PfCRT-mediated CQ transport is from the lumen of the digestive vacuole with a pH of 5.2 to the cytoplasm with a pH of ~ 7.2. Both the N- and C-terminus are oriented towards the digestive vacuolar lumen (Kuhn et al., 2007). The orientation of E207 in the open-to vacuole conformation was taken from published structural data based on single particle imaging [Kim, 2019 #21]. We have included the following statement:

“The direction of radio-labeled drug transport is here from the mostly acidic extracellular solution (pH 4.5 or 6.0, as indicated) into the oocyte cytosol (pH ~ 7.2), which corresponds to the efflux of drug from the acidic digestive vacuole (pH 5.2) into the parasite cytoplasm (pH 7.2). The topology of PfCRT in the oolemma is such that both the N- and C-terminus face the extracellular milieu^{3,22}. In the parasite, the N- and C-terminus face the lumen of the digestive vacuole²³.” (Methods, page 21)

4) Given that kinetic measurements of E207A have been made, could the authors apply the same model selection process to those data with the same model selection criterion? This process should select a model without competition. This approach could be used to validate the general approach of model selection.

We did kinetic measurements of E207A, however, only at two pH values – pH 4.5 and 6.0. No pH effects were observed, meaning, the kinetics at pH 4.5 is not statistically different from that conducted at pH 6.0. We, therefore, did not extend the kinetics study to other pH values. Subjecting the available two kinetics measurements of E207H to the statistical model analysis did

not produce any meaningful results, which is expected since the pH had no effect on the CQ transport kinetics of E207.

5) "E207 is not involved in proton transfer pathway" -- It's not clear to me that proton-driven transporters require proton-transfer chains. At least looking at secondary active transporters such as MFS transporters (eg LacY) and sodium/proton coupled transporters (eg TtNapA, PaNhaP), a single titratable residue that changes accessibility in the alternating access conformational transition may be sufficient; in other transporters (such as PiPT, MATE) two-step proton relay may be occurring. If the alternating access mechanism exposes the binding residue to solvent, transport chains for proton hops are not strictly necessary. Thus, although I agree with the authors' hypothesis that E207 is not directly involved in proton transfer, my reasoning would rather be that it is at the periphery and not at the center of the binding site around which the alternating access transition pivots.

We fully agree with the reviewer. Our initial text was misleading. We extended our investigation on the proton transfer mechanism and now provide initial evidence implicating D137 in this process. This additional work has led to a new section in the results (results, page 8 to 9 and new Supplementary Figure 12). We have further amended the discussion as following:

"We also found no evidence of E207 partaking in a putative proton transfer. The peripheral position of E207 would require an elaborated pathway to cross the pore. Although E207 is involved in hydrogen bond networks, these hydrogen bond networks involve the immediate surroundings of E207 and do not extend all the way from one site of the transporter to the other. Furthermore, some of the E207 hydrogen bond networks also included interactions with the protein backbone, which would be unusual for a proton transfer pathway (Supplementary Fig. 11).

A more promising candidate for proton binding and transfer is residue D137 on TM 3. Inserting a negatively charged residue in a transmembrane domain is energetically unfavorable ($\sim 4\text{--}5$ kcal mol⁻¹) and is, therefore, avoided unless the residue plays a functional role, e.g., in proton-transfer⁴⁶. To couple protonation dynamics with conformational changes, negatively charged residues frequently engage in inter-helical hydrogen bond formation via their carboxylate group with the hydroxyl group of Ser or Thr⁴⁶. This particular inter-helical carboxyl-hydroxyl interaction appears to be a hallmark of a proton-transfer mechanism in a membrane transporter⁴⁶. MD simulations indicated that D137 forms inter-helical hydrogen bonds with S227 on TM 6 in both the open-to-vacuole and occluded conformations, supporting its involvement in proton transfer (Supplementary Fig. 12). Additionally, substitution of D137 with the isosteric amino acid Asn greatly reduced CQ transport activity, consistent with the role of D137 in proton shuttling (Fig. 1d). While initially considering D329 as a potential candidate for proton transfer, we favored an alternative function due to its absence of inter-helical hydrogen bonding with Ser or Thr, such as an involvement in substrate binding³⁶. Further experiments, including constant pH MD simulations, are necessary to fully elucidate the functions of both residues in the transport cycle of PfCRT^{Dd2}." (Discussion, page 17)

6) In the discussion of the interaction network it was not entirely clear if the distances and networks were only computed from static models (from experiment or modeling). If only static models were used then they may contain spurious interactions. By basing such discussions on MD data one can appreciate the uncertainty better: Show distributions of the discussed distances and for simulations with different protonation states explicitly modeled (in particular E207 (see above), potentially also for E372 and K80).

In the previous version of the manuscript, interaction networks were presented for static models only. Following the reviewer's suggestion, in the revised manuscript, we calculated the distances between hydrogen bond donors and acceptors in residues E207, K80, E372 and N84 in simulations with different conformations of PfCRT (open-to-vacuole, occluded, open-to-cytoplasm) and different protonation states for E207. The distances between hydrogen bond donors and acceptors is presented in the new Figure 5 and the new Supplementary Figure 14). Histograms showing the distributions of the distances are shown in (Supplementary Figure 15). In the simulations we could observe the formation of hydrogen bonds between E207 and K80 in the occluded and open-to-cytoplasm conformations only when E207 is in the deprotonated state.

7) The authors present good evidence for the importance of the E207-K80 salt bridge. As an additional key validation I suggest the charge-swap double mutation E207K/K80E. If the salt-bridge is important, this may restore close to WT transport.

We thank the reviewer for this wonderful idea. As suggested by the reviewer, we have generated the E207K/K80E double mutant. The mutant was pH-sensitive and fully CQ restored transport activity at pH 6.0. The new results are shown in the amended Fig. 6a and are described in the results and discussion, as follows:

"These findings suggest a critical interaction between E207 and K80 during transition from the open-to-vacuole to the occluded conformation. To investigate this conclusion further, we generated a double mutant in which we swapped the two residues. The resulting E207K/K80E double mutant was pH-sensitive (R-value of 0.54 ± 0.18) and, importantly, had a transport activity at pH 6.0 comparable to that of PfCRT^{Dd2} ($4.75 \pm 0.91 \text{ pmol h}^{-1} \text{ oocyte}^{-1}$) (Fig. 6a)." (Results, page 13)

"The importance of the interaction between E207 and K80 is further supported by the E207K/K80E charge-swap variant that displayed a restored pH-sensitivity and a fully restored transport activity at pH 6.0 (Fig. 6a)." (Discussion, page 18)

8) Line 287 "Having a pKa of 6.0, His can also be protonated".

This is fallacious reasoning because it invokes the solution pKa of His. The protein micro environment can modify the pKa. Without providing evidence that the actual pKa is ~6, this argument does not contribute anything to the discussion. You could possibly create a mutant model and use PROPKA pKa predictions (similar to Suppl. Table 2) to get an idea of the pKa of E207H.

Following the reviewer's suggestion, we simulated the mutant E207H for the three conformations of PfCRT, open-to-vacuole, occluded and open-to-cytoplasm. For each conformation, three replicas of 200 ns were performed. We then collected 45 protein structures from the simulations and predicted pKa values for them. The range of pKa values is around 6 for each conformation (pKa values of 5.65-7.06, 4.20-7.81 and 5.95-7.27 for the open-to-vacuole, occluded and open-to-cytoplasm conformations, respectively). However, we did not observe any specific interactions with E207H. We mention this finding in the results section:

“Although MD simulations of the E207H mutant did not detect hydrogen bond formation between the His imidazole side chain and K80 or any other residue, this may be due to the lack of electrostatic attraction and, consequently, of a weaker interaction.” (Results page 14)

The estimated pKa values of E207A in the different conformations are shown below for review purposes.

Table 2. Average, standard deviation (STD), minimum (min) and maximum (max) pKa values predicted for H207 in the mutant E207H of PfCRT^{Dd2}. Protein configurations for calculations were obtained from three independent MD simulations of the mutant E207H, in different configurations (open-to-vacuole, occluded, open-to-cytoplasm). Predictions were done with Propka 3.5.

residue	average	STD	Min	Max
open-to-vacuole	6.44	0.32	5.65	7.06
occluded	6.06	0.65	4.20	7.81
open-to-cytoplasm	6.69	0.42	5.95	7.27

9) If only a hydrogen bond is important and not a salt-bridge then shouldn't E207Q should have same transport but no pH dependence? Was this mutant measured? If E207Q had similar transport properties as WT but avoided the pH sensitivity, wouldn't this be a favorable mutant?

The mutant is shown in the manuscript (see Fig. 6 and new Supplementary Table 5). The mutant is largely pH-insensitive, with an R-value of 0.88 ± 0.01 , as predicted by the reviewer. However, the transport activity at pH 6.0 is only half of that of PfCRT^{Dd2} (2.4 ± 0.5 versus 5.1 ± 0.2), which we explain by the critical role the salt bridge between E207 and K80 has in accelerating the transport cycle. We have compiled the R-values and transport activities at pH for all E207 replacement mutants in the new Supplementary Table 5.

10) " $1 > \Delta AICc < 7$ " is very confusing! suggest to express in words what is meant, e.g. "Models with $\Delta AICc$ between 1 and 7." or " $1 < \Delta AICc < 7$ ".

We apologize for the confusion and have amended the text as suggested by the reviewer.

11) AIC was used to choose the best model but no model quality check of the best model itself was provided. E.g. are the residuals (RSS) Gaussian distributed, could cross-validation be performed (split data in training/test set)?

We apologize for the oversight and now present the requested data in the new Supplementary Fig. 10. The findings are now described in the results section:

“The quality of the partial non-competitive inhibition model was checked by superimposing the experimentally derived data points with the theoretical expectations in a 3D plot (Supplementary Fig. 10a). To assess the goodness of the fit, we determined the residuals (difference between the observed and expected uptake values) and plotted them as a function of the CQ and proton concentration (Supplementary Fig. 10b)⁴¹. The residuals were found to be randomly distributed across the x axis within each CQ and proton concentration, indicating that the partial non-competitive inhibition model can account for the interaction of PfCRT^{Dd2} with CQ and protons.” (Results, page 10)

12) PROPKA is heuristic and indicative of the chemical environment of a residue but very dependent on detailed conformations. Running PROPKA on frames of MD trajectories can provide an indication of the variability and the distribution of pKa values.

Following the reviewer’s suggestion, in the revised version of the manuscript, we performed 3 200-ns MD simulations of the three different conformations of PfCRT (open-to-vacuole, occluded, open-to-cytoplasm) and two different protonation states for E207. We used 45 structures from simulations for pKa calculations. The pKa values estimated for E207 tend to be lower for the occluded conformation, with E207 deprotonated (pKa range 2.34-5.27, Supplementary Table 4), in comparison with the occluded conformation, with E207 protonated (pKa range 4.19-6.78, Supplementary Table 4), the open-to-vacuole (pKa ranges of 3.75-5.18 and 3.21-5.60 for E207 protonated and deprotonated, respectively, Supplementary Table 4) and open-to-cytoplasm conformations (pKa ranges of 3.52-5.33 and 2.81-5.53 for E207 protonated and deprotonated, respectively, Supplementary Table 4).

13) Note that D329 also shows a substantial up-shift in pKa (5.2 to 9.0) according to PROPKA. If E207 is discussed based on the PROPKA results then the authors should also comment on D329 and discuss why (or why not) it may be relevant. Otherwise it appears as if results are being cherry-picked.

Inspired by your comment and that of reviewer 1, we have investigated a possible function of D329 in the proton transfer mechanism. However, the results obtained do not favor such a function. Willems et al. (2023) recently suggested a role of D329 in CQ binding, which would agree with unpublished work from our lab implicating D329 in substrate recognition. We have included new sections on proton transfer in the results and discussion:

“To identify more plausible candidates for proton-transfer, we re-examined the alanine mutants, but this time with a different set of criteria in mind. As shown in other systems, residues involved in proton-binding and proton-transfer are titratable and frequently consist of Glu and Asp; they are present in a transmembrane domain and are located in the center of the channel; they are able to engage in inter-helical H-bonding, frequently with Ser or Thr, for coupling protonation dynamics with conformational changes; and replacing them strongly affects the transport activity⁴⁵⁻⁴⁷. Applying these criteria identified D137 and D329 as initial candidates. Both residues are located in transmembrane domains (TM3 and TM8, respectively; Supplementary Fig. 2) and replacing them by the isosteric amino acid Asn drastically reduced the transport activity for CQ at pH 6.0 (Fig. 1d). Both residues are conserved among *Plasmodium* spp (Supplementary Fig. 4a) and structural models placed both residues in the center of the channel (Supplementary Fig. 12). MD simulations revealed that D137 forms an inter-helical hydrogen bond with S227 (residing on TM 6; distance of 1.85 Å) in the open-to-vacuole and the occluded conformation (Supplementary Figs. 12a and b, and 13), consistent with a previous report³⁶. In comparison, D329 engages in intra-

helical hydrogen bonding - with T333 (like D329 also residing on TM 8; distance of 2.26 Å) (Supplementary Figs. 12c and d, and 13). These initial findings would favor D137 as a plausible candidate for proton shuttling, although further experiments are needed to clarify the role of D137 and also that of D329 in the transport cycle.” (Results page 11 and 12)

“A more promising candidate for proton binding and transfer is residue D137 on TM 3. Inserting a negatively charged residue in a transmembrane domain is energetically unfavorable ($\sim 4\text{--}5$ kcal mol⁻¹) and is, therefore, avoided unless the residue plays a functional role, e.g., in proton-transfer⁴⁵. To couple protonation dynamics with conformational changes, negatively charged residues frequently engage in inter-helical hydrogen bond formation via their carboxylate group with the hydroxyl group of Ser or Thr⁴⁵. This particular inter-helical carboxyl-hydroxyl interaction appears to be a hallmark of a proton-transfer mechanism in a membrane transporter⁴⁵. MD simulations indicated that D137 forms inter-helical hydrogen bonds with S227 on TM 6 in both the open-to-vacuole and occluded conformations, supporting its involvement in proton transfer (Supplementary Fig. 12). Additionally, substitution of D137 with the isosteric amino acid Asn greatly reduced CQ transport activity, consistent with the role of D137 in proton shuttling (Fig. 1d). While initially considering D329 as a potential candidate for proton transfer, we favored an alternative function due to its absence of inter-helical hydrogen bonding with Ser or Thr, such as an involvement in substrate binding³⁶. Further experiments, including constant pH MD simulations, are necessary to fully elucidate the functions of both residues in the transport cycle of PfCRT^{Dd2}.” (discussion, page 19).

14) Provide some minimal quality assessment for structural models: - CA RMSD (all residues and only transmembrane domains) - secondary structure (e.g. DSSP), either per-residue time series or probability to maintain secondary structure

Following the reviewer’s suggestion, we provide a quality assessment (RMSD of all alpha-carbons, RMSD of the alpha-carbons of the transmembrane regions, and DSSP analysis to assess the preservation of secondary structure) for the simulations of the three conformations of PfCRT (new Supplementary Figures 8 and 9). Additionally, the new Supplementary Table 3 provides quality assessment for the structural models built for the three conformations of PfCRT.

15) For reproducibility and transparency, make structural models of PfCRT publicly available, namely (1) open-to-cytoplasm and (2) occluded conformation. PDB files can be added to Supplementary Information or (better) in a public archive (with DOI) such as zenodo or figshare.

The structural models built for the three conformations are available in zenodo, in the link below: <https://doi.org/10.5281/zenodo.7828870>. Additionally, we also provided the systems used to start the MD simulations for each conformation (with E207 in the deprotonated state). We have included the information in the data availability statement:

“**Data availability.** The authors declare that the data supporting the findings of this study are available within the article and its supplementary information files, or available from the authors upon request. The original data and the structural models of PfCRT^{Dd2} underlying this article are available via the Dryad Digital Repository at <https://doi:10.5061/dryad.2z34tmprm> (for review purposes:

<https://datadryad.org/stash/share/4VDzilAC5UaISwDyi3q1Zf6cnrWcYKDaHzqdjsruGEo>) and the Zenodo Digital Repository at <https://doi.org/10.5281/zenodo.7828870>, respectively.” (page 32)

16) Add more details to modeling methods:

- For setting of protonation states at pH 6 PDB2PQR was used but state in main paper that really PROPKA was used (at the moment, this is only clear by reading Suppl Table 2) and cite PROPKA.

We apologize for the confusion in the description of the program used to predict pKa values and protonation states. We used the program Propka version 3.5, as implemented in the program pdb2pqr version 2.1.1, to predict pKa values and assign protonation states. The description of the programs used is modified in the Methods section in the revised version of the manuscript, and the appropriate references were added.

“Prior to membrane embedding of the protein, the protonation states of the residues at pH 6 were determined using Propka version 3.5⁶³⁻⁶⁵, as implemented in the program pdb2pqr version 2.1.1^{66,67}.” (Methods, page 25)

- add citations for thermostat and barostat algorithms, LINCS, SETTLE (for water)

The citations were added in the revised version of the manuscript.

- state treatment of non-bonded interactions (Coulomb and van der Waals/LJ) (with citation for eg Smooth PME)

The explanation of the treatment for non-bonded interactions and the appropriate citations were added in the revised version of the manuscript.

17) Fig 1: sub-labels a-e are missing; Fig 4: sub-labels a-d are missing

We apologize for the oversight and have corrected the figures.

Additional references

- Bakouh N, Bellanca S, Nyboer B, Moliner Cubel S, Karim Z, Sanchez CP, Stein WD, Planelles G, Lanzer M (2017) Iron is a substrate of the Plasmodium falciparum chloroquine resistance transporter PfCRT in Xenopus oocytes. *J Biol Chem* 292: 16109-16121
- Bellanca S, Summers RL, Meyrath M, Dave A, Nash MN, Dittmer M, Sanchez CP, Stein WD, Martin RE, Lanzer M (2014) Multiple drugs compete for transport via the Plasmodium falciparum chloroquine resistance transporter at distinct but interdependent sites. *J Biol Chem* 289: 36336-36351
- Bozzi AT, McCabe AL, Barnett BC, Gaudet R (2020) Transmembrane helix 6b links proton and metal release pathways and drives conformational change in an Nramp-family transition metal transporter. *J Biol Chem* 295: 1212-1224
- Coppee R, Sabbagh A, Clain J (2020) Structural and evolutionary analyses of the Plasmodium falciparum chloroquine resistance transporter. *Sci Rep* 10: 4842
- Forrest LR (2013) Structural biology. (Pseudo-)symmetrical transport. *Science* 339: 399-401
- Gunshin H, Mackenzie B, Berger UV, Gunshin Y, Romero MF, Boron WF, Nussberger S, Gollan JL, Hediger MA (1997) Cloning and characterization of a mammalian proton-coupled metal-ion transporter. *Nature* 388: 482-488
- Kim J, Tan YZ, Wicht KJ, Erramilli SK, Dhingra SK, Okombo J, Vendome J, Hagenah LM, Giacometti SI, Warren AL, Nosol K, Roepe PD, Potter CS, Carragher B, Kossiakoff AA, Quick M, Fidock DA, Mancia F (2019) Structure and drug resistance of the Plasmodium falciparum transporter PfCRT. *Nature* 576: 315-320
- Kuhn Y, Rohrbach P, Lanzer M (2007) Quantitative pH measurements in Plasmodium falciparum-infected erythrocytes using pHluorin. *Cell Microbiol* 9: 1004-1013
- Liao J, Li H, Zeng W, Sauer DB, Belmares R, Jiang Y (2012) Structural insight into the ion-exchange mechanism of the sodium/calcium exchanger. *Science* 335: 686-690
- Martin RE, Marchetti RV, Cowan AI, Howitt SM, Broer S, Kirk K (2009) Chloroquine transport via the malaria parasite's chloroquine resistance transporter. *Science* 325: 1680-1682
- Parker JL, Li C, Brinth A, Wang Z, Vogeley L, Solcan N, Ledderboge-Vucinic G, Swanson JMJ, Caffrey M, Voth GA, Newstead S (2017) Proton movement and coupling in the POT family of peptide transporters. *Proc Natl Acad Sci U S A* 114: 13182-13187
- Shafik SH, Cobbold SA, Barkat K, Richards SN, Lancaster NS, Llinás M, Hogg SJ, Summers RL, McConville MJ, Martin RE (2020) The natural function of the malaria parasite's chloroquine resistance transporter. *Nat Commun* 11: 3922
- Summers RL, Dave A, Dolstra TJ, Bellanca S, Marchetti RV, Nash MN, Richards SN, Goh V, Schenk RL, Stein WD, Kirk K, Sanchez CP, Lanzer M, Martin RE (2014) Diverse mutational

pathways converge on saturable chloroquine transport via the malaria parasite's chloroquine resistance transporter. *Proc Natl Acad Sci U S A* 111: E1759-1767

Willems A, Kalaw A, Ecer A, Kotwal A, Roepe LD, Roepe PD (2023) Structures of Plasmodium falciparum Chloroquine Resistance Transporter (PfCRT) isoforms and their interactions with chloroquine. *Biochemistry* 62: 1093-1110

Annex 1

Table 1. Average, standard deviation (STD), minimum (min) and maximum (max) pKa values predicted for titratable residues of PfCRT in top 10 models of the open-to-cytoplasm conformation.

Residue	Average	STD	min	max
ASP137	7.635	0.78	6.65	9.09
ASP241	3.872	0.37	3.25	4.17
ASP310	4.156	0.10	3.97	4.33
ASP311	5.777	0.37	5.36	6.41
ASP313	3.689	0.40	3.18	3.91
ASP329	4.749	0.33	4.25	5.17
ASP338	8.032	2.14	5.35	9.44
GLU75	6.658	1.85	3.59	8.1
GLU95	8.499	0.29	8.01	8.72
GLU121	4.923	0.85	4.48	6.65
GLU198	6.782	1.07	6.11	9.44
GLU204	4.286	0.26	3.96	4.5
GLU207	5.857	0.31	5.45	6.08
GLU208	5.375	0.47	4.72	5.74
GLU232	5.427	1.39	4.3	7.89
GLU271	4.862	0.13	4.79	5.02
GLU278	4.093	0.26	3.81	4.7
GLU299	4.374	0.23	3.97	4.63
HIS97	4.37	1.59	3.06	6.43
HIS123	6.012	0.79	3.78	6.39
HIS180	6.129	0.17	5.79	6.27
HIS273	6.221	0.03	6.18	6.26
CYS72	12.461	0.49	12.04	12.82
CYS101	13.955	1.28	12.33	16.54
CYS139	10.101	0.95	9.64	12.78
CYS171	9.271	0.13	9.17	9.58

CYS225	11.051	0.47	10	11.65
CYS258	9.843	0.50	9.46	11.16
CYS289	99.99	0.00	99.99	99.99
CYS301	99.99	0.00	99.99	99.99
CYS309	99.99	0.00	99.99	99.99
CYS312	99.99	0.00	99.99	99.99
CYS328	11.22	0.50	10.35	11.79
CYS350	11.442	0.34	11.07	11.97
TYR62	9.663	3.14	0.76	10.99
TYR68	11.96	0.68	10.82	12.64
TYR89	11.932	0.98	10.72	12.97
TYR109	10.24	0.20	10.09	10.48
TYR177	10.027	0.18	9.55	10.27
TYR179	10.357	0.38	10.03	11.04
TYR182	10.55	0.66	10.11	12.36
TYR184	15.945	1.84	13.6	18.1
TYR238	11.589	2.06	9.76	13.85
TYR264	12.382	0.51	11.8	12.89
TYR276	10.073	0.01	10.06	10.08
TYR335	10.703	0.84	9.91	12.33
TYR345	10.482	0.87	10.04	12.87
TYR360	13.014	0.80	11.49	13.96
LYS80	6.509	1.84	3.13	8.8
LYS85	10.122	0.21	9.68	10.38
LYS115	10.393	0.11	10.14	10.48
LYS116	10.364	0.18	10.33	10.47
LYS120	10.36	0.14	10.08	10.46
LYS200	10.592	0.17	10.3	10.85
LYS236	9.068	0.71	7.89	9.97
LYS237	9.849	1.13	6.86	10.43
LYS239	9.596	0.93	8.84	10.17
LYS270	10.436	0.03	10.39	10.49

LYS284	9.853	0.32	9.39	10.09
LYS307	11.211	0.20	10.82	11.51
LYS317	10.374	0.04	10.32	10.41
LYS339	9.367	1.49	6.15	10.55
ARG81	11.632	0.93	10.71	12.63
ARG122	12.175	0.38	11.19	12.45
ARG124	11.73	1.03	9.47	12.4
ARG150	12.055	0.19	11.97	12.27
ARG176	12.349	0.13	12.06	12.57
ARG178	12.364	0.11	12.31	12.47
ARG231	9.046	0.58	8.26	9.88
ARG244	12.605	0.17	12.35	12.8
ARG294	12.314	0.02	12.27	12.34

REVIEWERS' COMMENTS

Reviewer #2 (Remarks to the Author):

All my comments have been addressed satisfactorily...I have no further comments.

Reviewer #3 (Remarks to the Author):

In the revised manuscript, the authors have considerably improved the manuscript and they have answered most of my questions.

From the revised manuscript and their responses, there is one issue and it is about using the triose-phosphate /phosphate translocator (5Y79) as the template for modeling of the occluded state. The sequence similarity is 15% which is very low (in a very lower-bound threshold of the twilight zone). How confident the authors are about their sequence alignment and their model? and with this low homology, wouldn't be better to use the variant of AlphaFold to sample the different conformations?

Reviewer #4 (Remarks to the Author):

The authors addressed my comments. In particular, they performed many additional MD simulations in triplicate (200 ns each) for different conformations and protonation states of the key residue E207. They show quality measures that indicate that their models behave well in simulations. Overall, their analysis now fully supports their conclusions, especially the relevance of the E207-K80 interaction (saltbridge) that only occurs when E207 is deprotonated and is primarily visible in the intermediate occluded conformation.

I am happy that the authors tried the E207K/K80E charge swap mutant and found further strong validation for the necessity of a charge-charge interaction between residues at 207 and 80.

I like the new discussion of D137 and D329, which comments on the putative proton transport pathway and thus addresses an obvious question that readers will likely have after finding out that E207 is **not** a proton-transporting residue.

Thank you for making structural/simulation data available in a public repository.

Overall, the authors have done a lot of excellent additional work in response to my own (and the other reviewers') comments to make this a well-rounded and insightful manuscript.

Responses to reviewers' comments on manuscript NCOMMS-22-47067A

Reviewer #2 (Remarks to the Author):

All my comments have been addressed satisfactorily...I have no further comments.

We thank the reviewer for his/her support.

Reviewer #3 (Remarks to the Author):

In the revised manuscript, the authors have considerably improved the manuscript and they have answered most of my questions.

From the revised manuscript and their responses, there is one issue and it is about using the triose-phosphate /phosphate translocator (5Y79) as the template for modeling of the occluded state. The sequence similarity is 15% which is very low (in a very lower-bound threshold of the twilight zone). How confident the authors are about their sequence alignment and their model? and with this low homology, wouldn't be better to use the variant of AlphaFold to sample the different conformations?

We thank the reviewer for bringing up this issue. AlphaFold is indeed a revolutionary tool. We are also aware of a previous study by (Willems et al., *Biochemistry*, 62:1093-1110,2023; also cited in the manuscript) who simulated the open-to-vacuole structure of PfCRT using the AlphaFold platform. However, many roads lead to Rome. We opted to perform homology modeling using Phyre 2. In the homology modeling performed by Phyre2, every candidate template has an associated sequence similarity and a confidence score, which represents the probability (from 0 to 100%) that the match between the input sequence and the template is a true homology.

In the case of the sequence of PfCRT and the candidate template PDB 5Y79, there is 15% sequence similarity and a confidence score of 99.7%, indicating a high probability of homology between the template structure and PfCRT. According to the authors of Phyre2 (Kelley et al, *Nat Protoc*, 10: 845–858, 2015), if the confidence score is larger than 90%, modeling is feasible, even if the sequence similarity is low. Still according to the authors of Phyre2, for a confidence score larger than 90% and low sequence similarity, one can be very confident that the protein adopts an overall fold similar to the template structure, and that the core of the protein is modeled at high accuracy (2-4 Å RMSD from the true structure).

Moreover, PDB 5Y79 has been used as a template in a previous publication, in combination with Phyre2, to model the occluded state of PfCRT (Coppee et al, *Sci. Rep.*, 10: 4842, 2020).

We also tried to use AlphaFold 2 to model the occluded state of PfCRT, but it did provide satisfying results. In our multiple attempts, we obtained structures very similar to the open-to-vacuole conformation only. We further would like to emphasize that key predictions made by the simulations have been supported by experimental data in the manuscript.

In response to the reviewer's comment we have added the following statement to the Methods section:

“Despite a low sequence identity between PfCRT and the triose-phosphate/phosphate translocator, a high confidence score of 99.7 % was obtained for the modeled PfCRT structure. Confidence score larger than 90% indicate that the modeled protein adopts an overall folding similar to the template structure, and that the core of the protein is modeled at high accuracy (2-4 Å RMSD from the true structure) ^[58].” (Methods, page 24).